# Behavioral screening defines the molecular Parkinsonism-related subgroups in *Drosophila*

Natalie Kaempf [1,2,3], Jorge S. Valadas[1,2], Pieter Robberechts [4,5], Nils Schoovaerts[1,2], Roman Praschberger [1,2,3,6], Antonio Ortega [1,2], Eliana Nachman [1,2], Lorenzo Ghezzi[1,2], Ayse Kilic[1,2], Dries Chabot[1,2], Uli Pech[1,2], Sabine Kuenen [1,2], Sven Vilain [1,2,8], El-Sayed Baz [1,2,7], Jeevanjot Singh[1,2], Jesse Davis[4,5], Sha Liu [1,2] & Patrik Verstreken [1,2] ✉

Parkinson's disease (PD) and related familial Parkinsonism are defined by motor dysfunction, but the specific upstream molecular causes of these clinical symptoms can vary widely. We hypothesize that these causes converge onto a limited number of core cellular pathways. To investigate this, we created a collection of 24 genetically well-controlled *Drosophila* models of familial forms of PD and related mono-genic forms of Parkinsonism. Using unbiased behavioral screening and machine learning we identify clusters of mutants that converge on (1) mitochondrial function; (2) retromer/vesicle trafficking and proteostasis/autophagy. Genes within each cluster have a similar genetic interaction profile and compounds that target specific molecular pathways ameliorate dopaminergic neuron dysfunction in a cluster-specific manner. Together, our data indicate that familial PD and related forms of Parkinsonism may fall into two broad functional groups, and may inform further work toward targeted biomarker discovery and therapeutic development.

Parkinson's disease and related genetic causes of Parkinsonism (for simplicity referred to as "Parkinsonism" hereafter) are characterized by motor symptoms such as bradykinesia, rigidity and tremor, caused by the progressive dysfunction and loss of dopaminergic neurons in the *substantia nigra*. However, patients often report non-motor issues decades before the onset of these motor symptoms, including hyposmia, and various sleep disturbances. In some cases, patients may also experience dementia and epileptic seizures[1–5] (Supplementary Table 1). While therapies to restore dopamine tonus can relieve motor symptoms[6], they do not address non-motor problems and they do not stop disease progression[5,7,8]. These treatments can also lead to side effects including hallucinations, psychosis and agitation[9]. This underscores the need for developing disease-modifying strategies that are grounded in the biological mechanisms of Parkinsonism.

While there are common motor and non-motor symptoms in Parkinsonism, the clinical presentation may vary[10–13]. There is work using these variations in an attempt to define disease subtypes[14–16], but these strategies have met limitations. Factors contributing to this challenge include the inherent genetic heterogeneity among patients and the limited availability of prodromal and longitudinal quantitative data. Furthermore, these clinical classifications pose experimental difficulties for therapeutic development as they are not associated with specific genetic mutations and thus molecular dysfunction[17].

[1]VIB-KU Leuven Center for Brain & Disease Research, Leuven, Belgium. [2]KU Leuven, Department of Neurosciences, Leuven Brain Institute, Leuven, Belgium. [3]Aligning Science Across Parkinson's (ASAP) Collaborative Research Network, Chevy Chase, MD, USA. [4]KU Leuven, Department of Computer Science, Leuven, Belgium. [5]Leuven.AI – KU Leuven Institute for AI, Leuven, Belgium. [6]Medical University of Innsbruck, Institute of Human Genetics, Innsbruck, Austria. [7]Zoology Department, Faculty of Science, Suez Canal University, Ismailia, Egypt. [8]Present address: Istanbul Medipol University, Istanbul, Turkey. ✉e-mail: patrik.verstreken@kuleuven.be

An alternative approach to classify Parkinsonism involves exploring its genetic complexity, providing a pathway-based perspective for stratification. Over 20 genes have been linked to familial forms of Parkinsonism, and recent genome-wide association studies (GWAS) have identified a growing number of loci that modify the risk of Parkinsonism[18–21]. These genetic findings offer valuable insight into the molecular mechanisms that potentially drive the disease[22–24]. However, the diversity of the potentially affected pathways poses the question of how, or whether, these Parkinsonism-related genes converge on common molecular networks to result in the same clinical disease. These molecular dysfunctions could manifest before dopaminergic-dependent motor symptoms given that patients experience complex behavioral and functional disturbances in the prodromal phase[25]. Thus, stratifying Parkinsonism based on molecular dysfunction could have significant implications for clinical trials and personalized treatment approaches already at an early stage.

In this study, we established a collection of 24 genetic *Drosophila* models for Parkinsonism and leveraged these to identify distinct disease subgroups. Using high-throughput behavioral monitoring, machine learning and pattern recognition algorithms, we were able to classify individual mutants in an unbiased manner. Our analysis revealed two major groups, whose existence was confirmed through extensive genetic interactions observed within each group. These groups share common molecular pathways related to mitochondrial function, vesicle trafficking/retromer and autophagy/proteostasis, and we demonstrate that targeted small molecule treatments can be tailored to these specific subgroups. This behavioral screening approach thus enabled us to categorize Parkinsonism into two major non-overlapping molecular pathways, providing a framework for understanding the disease and developing targeted treatments.

## Results

### Parkinsonism fly collection recapitulates dopamine responsive, progressive motor impairments

Using CRISPR/Cas9-mediated gene editing we created a comprehensive collection of *Drosophila* loss-of-function Parkinsonism models. We identified the 24 fly homologs of genes known to be causative or conveying risk for Parkinsonism in humans, based on extensive literature research and clinical insights. We replaced the first common exon of possible transcripts by a attP-flanked $w^+$ reporter cassette in each of the genes (Supplementary Tables 1, 2, Supplementary Fig. 1a, b, using the donor plasmid pWhiteSTAR ("WS"), "Methods"). These knockouts (KO) were backcrossed to isogenic control flies for >10 generations, allowing for detailed comparisons between these models independent of genetic variation—for *park* we were unable to generate such a knock-out and thus we resorted to the established *park*[1/Δ21] mutant[26,27]. Most of the Parkinsonism genes we included are thought to cause Parkinsonism or increase risk of the disease by a loss-of-function mechanism with some exceptions; nonetheless, we included the analysis of KO flies only as a way of comparing the genetic factors to one another without confounding our analysis at this stage with the effects of specific pathogenic mutations. These KO flies are loss of function mutants as confirmed by RT-PCR (Supplementary Fig. 1c) and are collectively referred to as 'Parkinsonism mutants'.

To assess whether these Parkinsonism mutants display locomotion phenotypes, we used a well-established climbing assay (SING—startle-induced negative geotaxis). SING defects in Parkinsonism fly models have previously been shown to involve dopaminergic impairments[28–30]. The performance of young Parkinsonism KO flies is similar to controls, except for *gba*[KO] flies that already exhibit a reduced SING score at young age (Fig. 1a). When aged, we detect a significant decline in motor performance in 80% of the Parkinsonism mutants (Fig. 1b), consistent with age-dependent dopaminergic impairments. To test this, we labeled dopaminergic neurons with anti-TH, which allowed us to assess synaptic connectivity onto their downstream target, the mushroom body (MB). In line with the SING results, we find that young Parkinsonism KO flies have normal innervation onto MB compared to controls (measured by the area of positive TH labeling, normalized to MB area as a measure of individual mutant brain size, Fig. 1d, Supplementary Fig. 2a) except for *gba*[KO] flies. This TH-labeled innervation area progressively declines (Supplementary Fig. 2b) and ultimately 90% of the Parkinsonism mutants with motor defects also show a significant dopaminergic neuron innervation defect (Fig. 1c, d and Supplementary Fig. 2c–f). We find that impairments in dopaminergic innervation onto MB significantly correlate with the defects in SING motor performance across our Parkinsonism mutants; $R^2$: 0.53 (Fig. 1e).

We next tested if Parkinsonism mutant flies with a progressive motor defect ($p \leq 0.001$) are L-Dopa responsive[6,31]. The motoric defects of most Parkinsonism mutants are significantly alleviated when flies are fed L-Dopa in the 10 days prior to the SING assay. We do not observe such rescue when flies are fed the stereoisomer D-Dopa (Fig. 1f) or $H_2O$ (solvent control, for selected mutants, Supplementary Fig. 2g)[27,32]. Thus, the majority of our Parkinsonism mutants suffer from a progressive L-Dopa-responsive motor defect.

In addition to typical motor symptoms, Parkinsonism patients sometimes experience additional progressive neurological defects, including epileptic seizures[4] (Supplementary Table 1). Although seizure-like activity in flies does not reproduce the full pathophysiology of human epileptic seizures, it serves as an accessible model of hyperexcitability[33–36]. We assessed seizure-like behavior using an established protocol based on mechanical hyper-stimulation of sensory inputs and assessing seizure-like activity ("Methods", ref. 36). While controls are fine, eleven aged Parkinsonism mutants show such behavior (Supplementary Fig. 3a–c, Supplementary videos 1–6). While SING defects in these flies are rescued by L-Dopa, seizure-like activity is not (Supplementary Fig. 3e). We then compared the familial Parkinsonism cases reported to suffer from epileptic seizures to the fly mutants exhibiting seizure-like behavior: 7 of the 24 familial forms of Parkinsonism were described to suffer from seizures[37–41], and 6 of these 7 homologous fly mutant models also display seizure-like activity (Supplementary Fig. 3d). There are also 5 other fly models with seizure-like behavior where clinical assessment of patients has not (yet) reported epileptic behavior (Supplementary Fig. 3d).

Taken together, our collection of Parkinsonism mutants recapitulates hallmarks of the disease, including typical age-dependent and dopamine-dependent neurological defects and defective dopaminergic neuron innervation. Several of the models also show epileptic-like behavior.

### High-throughput behavior monitoring reveals Parkinsonism subgroups with distinct sleep and activity patterns

We used the Parkinsonism mutants to stratify the disease based on comparing complex behavioral phenotypes across our genetic models. We placed individual flies in in-house-built ethoscopes that enable automated video tracking of movements over multiple diurnal cycles (Fig. 2a)[42–44]. We measured day-time activity, but also multiple other parameters such as features relevant to sleep. Sleep is an evolutionary conserved behavior[45,46] and various aspects of the sleep cycle are affected in the prodrome of Parkinsonism such as insomnia, excessive day-time sleepiness, and other alterations[47,48]. We first calibrated our system using the *pigment-dispersing factor* (*pdf*) null mutant, which is a well-studied circadian-deficient fly line. As expected, we observe defects in evening and morning anticipation (Fig. 2bi, iii), reduced latency and increased sleep at night compared to control (Fig. 2bii), thus validating our experimental system[49].

We then screened the entire Parkinsonism fly collection and tested ≥69 individual flies per genotype at a young age (2–5 days post eclosion) before the onset of dopaminergic motoric defects (Fig. 1a), and analyzed their activity profiles comprising basic motor

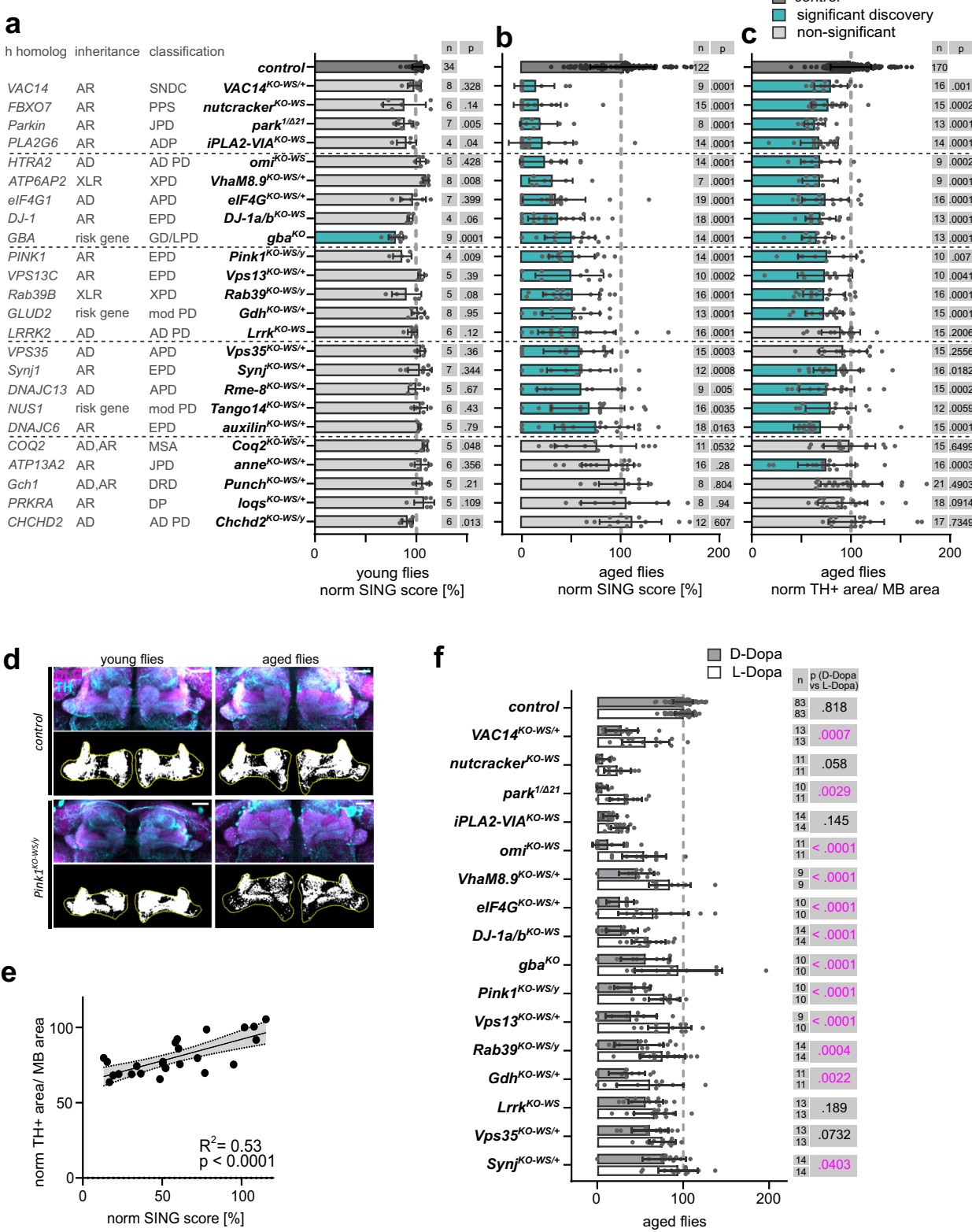

**d** young flies / aged flies — control, Pink1^KO-WS/y

**e** R² = 0.53, p < 0.0001 (norm SING score [%] vs norm TH+ area/ MB area)

**f** D-Dopa / L-Dopa — aged flies norm SING score [%]

performance, circadian rhythms (morning anticipation) and the sleep architecture (sleep fraction in light and dark periods; number and duration of sleep bouts; onset of longest bout) (Fig. 2a and Supplementary Fig. 4a–d), aspects of which are also frequently found affected in patients[2,50]. We find specific behavioral defects and none are common across all the mutants, indicating this type of analysis is capable of distinguishing specific features between Parkinsonism models

(Supplementary Fig. 4). Furthermore, we also show that these defects are not the result of basic motor performance impairment (e.g., velocity if awake) as this feature does not generally correlate with the behavioral parameters we quantified (Supplementary Fig. 5).

To analyze these results also in an unbiased and holistic manner, we employed unsupervised pattern recognition analysis, enabling us to decompose the ethoscope data of individual flies. Using non-

**Fig. 1 | Parkinsonism mutants show age- and dopamine-dependent phenotypes.** *Related to Supplementary Figs. 1–3 and Supplementary Tables* 1, 2.
**a**, **b** Quantification of SING at $6 \pm 1$ d (young, **a**) and $42 \pm 2$ d after eclosion (aged, **b**) of Parkinsonism mutants normalized (norm) to the control ($CS^{w1118}w^+$, dark gray). Due to their decreased survival *nutcracker*$^{KO-WS}$ were tested at 15 d, *Pink1*$^{KO-WS/y}$ at $22 \pm 2$ d, *iPLA2-VIA*$^{KO-WS}$ and *gba*$^{KO}$ mutants at $25 \pm 1$ d and included in the "aged" analysis. "/+" refers to heterozygosity and "/y" to hemizygosity. ANOVA Kruskal–Wallis with Benjamini–Hochberg. Bars: mean $\pm$ SD; points represent groups of animals, $n$ and $p$ values are indicated. **c** Quantification of dopaminergic synaptic area within MB area in aged mutants (as in (**b**)) relative to the control. ANOVA Kruskal–Wallis with Benjamini–Hochberg. Bars: mean $\pm$ SD; points are individual animals, $n$ and $p$ values are indicated. **d** Maximum projection confocal image of *control* and *Pink1*$^{KO-WS/y}$ fly brains (young: $5 \pm 1$ d and aged: $22 \pm 2$ d) stained with anti-TH (cyan) and anti-DLG (magenta) antibodies. The black-white images: thresholded TH area (white) of middle z-plane within the MB (outline in yellow). Scale bar: 20 μm. Representative images of Fig. 1c and Supplemantary Fig. 2a, *n*

indicated in graphs. **e** Linear regression plot with 95% confidence intervals (gray area) of the mean normalized SING score and TH positive area within MB area of aged Parkinsonism mutants show positive correlation with $R^2 = 0.53$, $p = 0.0001$. **f** SING quantification of aged flies (as in **b**) treated with L-Dopa or D-Dopa (control, gray bars) 10 d prior to the assay relative to D-Dopa treated control, Two-way ANOVA with Tukey's multiple comparison, $n$ and $p$ values are indicated in the graph. Bars: mean $\pm$ SD; points represent groups of animals. Source data are provided as a Source data file. ADP adult-onset dystonia-Parkinsonism, AD autosomal dominant, APD adult-onset Parkinson's disease, AR autosomal recessive, DLG Discs-large, DP dystonia-Parkinsonism, DRD DOPA-responsive dystonia, EPD early-onset Parkinson's disease, GD Gaucher disease, h human, JPD juvenile-onset Parkinson's disease, MB mushroom body, mod PD modifier for Parkinson's disease, MSA Multiple system atrophy, PPS parkinsonian-pyramidal syndrome, SING startle-induced negative geotaxis, SNDC childhood-onset striatonigral degeneration, TH tyrosine hydroxylase, WS white-STAR, XLR X-linked recessive, XPD X-linked Parkinsonism.

negative matrix factorization (NMF)[51,52] we identified distinct behavioral components (Fig. 2c, d). The elbow method and Calinski-Harabasz Index showed that 5 components explained most of the biological meaningful variance (Supplementary Fig. 4e, f, "Methods")[51,53]: 'Component 0' is defined by a sleep bout in the morning; 'Component 1' delineates increased sleep as the night progresses, especially in the early morning; 'Component 2' constitutes sleep late during the day; 'Component 3' is defined by a normal sleep pattern and 'Component 4' constitutes a strong sleep bout right at the start of the night. Each individual fly is represented by multiple behavioral components. For example, control flies are represented mostly by component 3 (Supplementary Fig. 4g–k). In contrast, *omi*$^{KO-WS}$ is mostly represented by component 1, and less by components 0 and 3, while *Pink1*$^{KO-WS/y}$ and *park*$^{1/\Delta21}$ flies are characterized chiefly by components 0, 2 and 4, but also to a lesser extent by the other components (Supplementary Fig. 4g–k). The mean principal component value of these features that represent each fly per genotype provides an unbiased quantification of the behavioral phenotype for each model. Also in these analyses, the components do not generally correlate with basic motor performance defects and with the behavioral parameters (Supplementary Figs. 5, 6, 7), meaning they can be treated as independent variables.

We then combined the unsupervised NMF-defined phenotypes with the explicit behavioral features and used hierarchical clustering to identify subgroups characterized by similarities in behavioral features (Fig. 2e–g). This delineates 2 major groups, each comprising 3 and 2 subgroups, respectively. Providing credence to our approach, *Pink1*$^{KO-WS/y}$ and *park*$^{1/\Delta21}$, whose protein products are known to interact to regulate mitophagy[54,55], cluster closely in group B1 (orange label). Furthermore, they also cluster together with another mitochondria-related gene *Chchd2*[56], suggesting behavioral clustering reveals common underlying biology. Furthermore, the adjacent group (B2, light orange) also includes genes involved in mitochondrial function such as *nutcracker (FBXO7)* and *Coq2*[57,58]. Among other elements, their behavior is characterized by increased sleep in the morning (component 0, Fig. 2e). The sleep traces of *park*$^{1/\Delta21}$ flies and *Pink1*$^{KO-WS/y}$ reveal a shortened latency to fall asleep (Fig. 2gi) and, as also previously reported, impaired anticipation of morning "lights-on" timing[59] (Fig. 2gii). On the other hand, *gba*$^{KO}$, *iPLA2-VIA*$^{KO-WS}$, *DJ-1a/b*$^{KO-WS}$, *Vps35*$^{KO-WS/+}$ and *Rab39*$^{KO-WS/y}$ flies cluster in 'group A1' (dark blue label) and show a reduced sleep fraction (Fig. 2fi). Similarly, the closest neighboring group (A2, medium blue label) also shows reduced sleep, especially during the day. Finally, group A3 (light blue label) comprises Parkinsonism mutants such as *VAC14*$^{KO-WS/+}$, *VhaM8.9*$^{KO-WS/+}$, *Lrrk*$^{KO-WS}$, *Rme-8*$^{KO-WS/+}$ and *eIF4G*$^{KO-WS/+}$ showing, e.g., reduced sleep late during the day (component 2) and increased numbers of sleep bouts per night.

We verified the existence of these groups using an independent methodology: principal component analysis (PCA). This reveals similar

groups of Parkinsonism-gene KO animals (Supplementary Fig. 8, groups outlined as in Fig. 2e) as those found using hierarchical clustering, except for *gba*$^{KO}$ that in the PCA associates with the genes in behavior group B (orange). This suggests the behavior of *gba* mutants shares characteristics of both groups A and B (Supplementary Fig. 8a). PC1 and PC2 together explain 58% of the total variance (Supplementary Fig. 8b) and are strongly influenced by individual behavior features (Supplementary Fig. 8c). PC1 is influenced by component 3 and 4 of our NMF analysis, the number of sleep bouts and the velocity if awake, while PC2 is mostly defined by sleep fraction, bout length and component 2 (Supplementary Fig. 8c). PC1 delineates the difference between the subgroups in groups A and B (light versus dark shades), while PC2 is most specific in delineating the difference between the mutants of groups A and B. Only the mutants in group A3 (light blue) are not discriminated well by our PCA. Hence, both types of unbiased analyses of the behavior parameters define 2 broad groups across 24 Parkinsonism mutants.

To assess the stability of the hierarchical clusters, we removed one behavioral feature or NMF component at a time from the original analysis (Supplementary Fig. 9a, b) and compared the newly generated groups (groups 1–5) to the originally determined clusters (behavior A1 - B2) using Jaccard similarity (Supplementary Fig. 9c–n), which computes the intersection-over-union[51]. This shows that in more than 50% of the cases, when one feature is removed, we find a Jaccard similarity higher than 0.35 (indicating substantial overlap[60–62]) (Supplementary Fig. 9o). This indicates the robustness of the subgroups we defined.

To test this further, we included the *pdf*$^{KO}$ mutant in our clustering. Circadian defects in *park* and *Pink1* mutant flies have been associated with impaired biogenesis of dense core vesicles containing the neuropeptide PDF[59]. Hence, we recorded the behavioral features of *pdf*$^{KO}$ animals, and also defined the components by NMF. *pdf*$^{KO}$ flies show behavioral similarities with *park*$^{1/\Delta21}$, *Pink1*$^{KO-WS/y}$ and *Chchd2*$^{KO-WS/y}$ mutants (green, Supplementary Fig. 10a, b) and, consistent with our prediction, *pdf*$^{KO}$ co-clusters with these mutants in behavior group B1 (orange, Supplementary Fig. 10b).

## Parkinsonism gene expression patterns do not correlate with subgroups

The convergence of Parkinsonism mutants into subgroups based on similar behavioral phenotypes could find its origin in shared molecular mechanisms or expression patterns across cell types of the genes in one group. To test the latter, we utilized single-cell sequencing datasets from our young control flies[30] and assessed the expression level of each of the fly Parkinsonism genes across all identified cell types in the brain (Supplementary Fig. 11a, b) and the entire body (Supplementary Fig. 11c). We then performed hierarchical clustering analysis on these gene expression profiles. For both the brain cell type dataset and the entire body cell dataset, this does not recapitulate the groups we

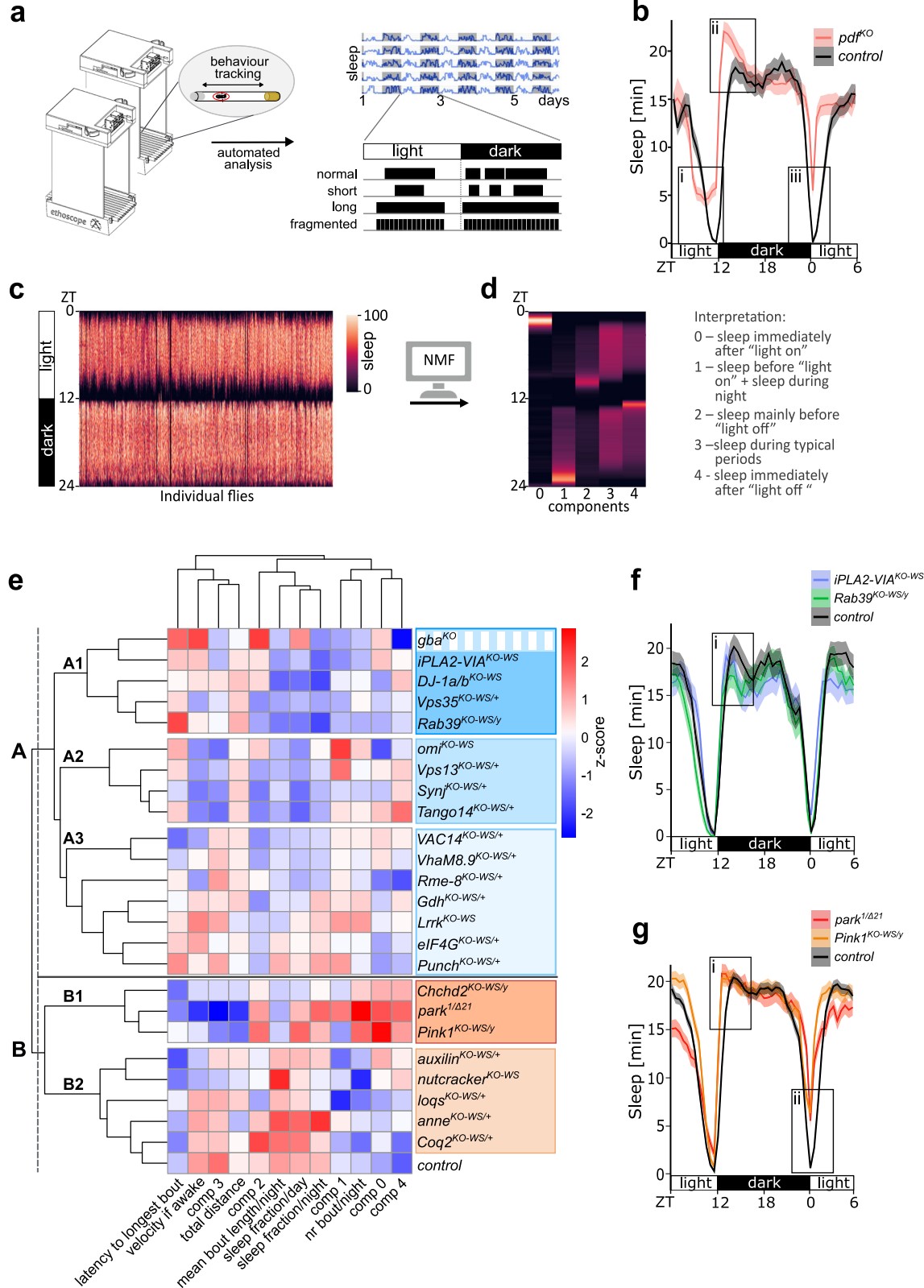

identified based on behavior. For example, the expression profiles of *Pink1*, *park* and *Chchd2* in the brain do not co-cluster and are expressed broadly (Supplementary Fig. 11a, b). Similarly, *gba1a/b*, *DJ-1a/b*, *Rab39*, *Vps35* and *iPLA2-VIA* that comprise group A1 do not exhibit similar or specific brain or body expression patterns or expression levels across

cell populations (Supplementary Fig. 11a–c). While we cannot exclude that genes could act within multicellular functional circuits comprising distinct cellular populations, we conclude that the clustering of genes based on behavior is not easily explained by shared expression profiles in the brain or body.

**Fig. 2 | Behavioral screening reveals 2 groups of Parkinsonism mutants.** *Related to Supplementary Figs. 4–10*. **a** Scheme of automated behavioral tracking and analysis using ethoscopes to describe complex activity profiles of young flies. Figure adapted with permission from Giorgio Gilestro[119]. **b** Average sleep traces of *pdf*$^{KO}$ (red, *n* = 71) and control (*CS*$^{w1118}$) (black, *n* = 71) showing advanced as well as reduced evening anticipation (**bi**), increased sleep (**bii**) and defective morning anticipation (**biii**). Solid lines and shaded areas indicate the mean and SEM of 5 light and dark cycles (12:12). **c** For pattern recognition analysis, behavior of individual flies averaged over 5 light and dark cycles (12:12) are shown as single lines. **d** Non-negative matrix factorization (NMF) decomposes the multivariant data into 5 biological meaningful components (0–4). **e** Hierarchical clustering of Parkinsonism mutants based on their scaled behavior data reveal 2 behavior groups (A in blue and B orange) with 3 and 2 subgroups, respectively (dark and light shades); *n* values are indicated in Supplementary Fig. 4k. *gba*$^{KO}$ marked in stripes as in PCA it clusters with group B (Supplementary Fig. 8). **f**, **g** Representative sleep traces of (**f**) *iPLA2-VIA*$^{KO-WS}$ (blue, *n* = 34), *Rab39*$^{KO-WS}$ (green, *n* = 36) and control (black, *n* = 42) showing (**fi**) reduced sleep and (**g**) *park*$^{1/\Delta21}$ (red, *n* = 69), *Pink1*$^{KO-WS/y}$ (orange, *n* = 98) and control (black, *n* = 226) showing (**gi**) increased sleep and shortened latency as well as (**gii**) decreased morning anticipation. Solid lines and shaded areas indicate the mean and SEM of 5 light and dark cycles (12:12). Source data are provided as a Source data file. Comp component, NMF non-negative matrix factorization, nr number, ZT zeitgeber.

## Genes in behavioral subgroups show similar genetic interaction profiles

We assessed the validity and robustness of the behavior-defined subgroups by performing genetic interaction (GI) analyses across all Parkinsonism mutants in vivo at a young age, similar to the age used in the behavior screening. In this independent approach, our strategy was to combine all mutants one-on-one with each other and to use a sensitive and quantitative electrophysiological readout of neuronal function: electroretinogram recordings (ERGs) that were found to be disrupted in many neurological disease models in flies. ERGs are the extracellular voltage response of the retina to a transient light pulse[63,64]. We analyzed the amplitude of the ERG-depolarization of single heterozygous mutants and the combinations of double trans-heterozygous Parkinsonism mutants (Fig. 3a, b). For *Pink1*$^{KO-WS/y}$, *Chchd2*$^{KO-WS/y}$ and *Rab39*$^{KO-WS/y}$ we used hemizygous flies, as their genes are located on the X chromosome. We then used a previously established methodology to define the GI as the difference between the prediction from a non-interacting model using a "multiplicative function" of 2 single heterozygous mutant flies and the observed value of the double heterozygous gene pair[65,66]. This allowed us to define the GI of gene pairs as "aggravating", "alleviating", or "not interacting". For example, within the 276 combinations we tested (Supplementary Data 1), we find the phenotype of double heterozygous mutants of *iPLA2-VIA* and *DJ-1a/b* that are co-clustering in behavior group A (Fig. 2e, blue), to be less severe (alleviating) as compared to either of the heterozygous mutants alone (Fig. 3c). Thus suggests that they act in the same molecular pathway. Similarly, we find an alleviating GI of *Rme-8* and *VhaM8.9* (Supplementary Fig. 12a). Combing mutants of *Vps35* and *omi* or *Pink1* and *nutcracker*, which co-cluster in behavior group A and B, respectively (Fig. 2e), aggravates the phenotype seen in the single heterozygous mutants (Fig. 3e and Supplementary Fig. 12b, Supplementary Data 1), highlighting a strong GI between these genes. The latter is consistent with *nutcracker's* function in mitophagy and its molecular interaction with the protein product of *Pink1*[57]. In contrast, we also find a gene pairs from behavior group A, such as *Gdh* and *Vps13*, that show the same ERG amplitude as predicted from single heterozygous mutants suggesting no molecular interaction (Fig. 3d).

When we perform hierarchical clustering on the GI profiles of the Parkinsonism mutants, we find 2 subgroups: GI 1 and 2 (Fig. 3f). Interestingly, 'GI 1' comprises 6 out of 8 genes that also co-cluster in behavior group B (Fig. 2e, orange), including *Pink1*, *park*, *Chchd2*, *nutcracker*, *anne* and *auxilin*. Furthermore, the majority of mutants that cluster in behavior group A (Fig. 2e, blue) are found in 'GI 2', including *gba*, *Synj*, *DJ-1a/b*, *iPLA2-VIA* and *Tango14* as well as *Rme-8*, *VhaM8.9*, *Lrrk*, *VAC14* and *eIF4G*. We quantified the similarity of the behavior clustering approach and the GI approach using the Jaccard similarity index. This confirms that behavior group A significantly overlaps with GI 2 and behavior group B with GI 1 (Fig. 3g). Thus, both the unbiased behavioral analyses and the GI profiles define similar Parkinsonism subgroups, which ultimately define shared molecular handles into different disease subtypes.

## Cluster-specific rescue of dopaminergic neuron defects

Next, we tested if the common age-dependent defects in dopaminergic neurons across the Parkinsonism mutants (Fig. 1 and Supplementary Fig. 2) have a cluster-specific pathomechanistic origin. We targeted pathways based on the proposed functions of the genes in a subgroup. We supplemented mutants of behavior group B that includes *Pink1*$^{KO-WS/y}$, *park*$^{1/\Delta21}$, *anne*$^{KO-WS/+}$, *auxilin*$^{KO-WS/+}$ and *nutcracker*$^{KO-WS}$ (Supplementary Fig. 13a), with Coenzyme Q10, an electron carrier of the mitochondrial respiratory chain previously shown to alleviate *pink1* phenotypes[67]. Indeed, the reduced TH positive area of these mutants is significantly rescued; likewise, the SING score of *Pink1*$^{KO-WS/y}$ and *nutcracker*$^{KO-WS}$ is also significantly improved by Q10 (*we did not test the other mutants*– Fig. 4a, b, Supplementary Fig. 13b). In contrast, Q10 does not rescue the reduced TH positive area of mutants in behavior group A1 and A2 (dark blue and medium blue) and the SING score of *iPLA2-VIA*$^{KO-WS}$ and *DJ-1a/b*$^{KO-WS}$ (Fig. 4a, b, Supplementary Fig. 13b), in line with the idea of group-specific rescue mechanisms.

We next targeted endo-lysosomal trafficking in behavior group A[68,69]. This group comprises the trafficking proteins Rab39, Synj1 and Vps13 and the retromer regulators Vps35 and iPLA2-VIA (Supplementary Fig. 13a). Defects caused by mutations in *iPLA2-VIA* were shown previously to be rescued by stabilizing the retromer complex with the chemical chaperone R55[70] and we show that also dopaminergic synaptic innervation defects and SING are rescued (Fig. 4c, d, Supplementary Fig. 13c). Similarly, the dopaminergic defects in other members of group A, such as *DJ-1a/b*$^{KO-WS}$, *gba*$^{KO}$, *Rab39*$^{KO-WS/y}$, *omi*$^{KO-WS}$, *Vps13*$^{KO-WS/+}$, *Synj*$^{KO-WS/+}$ and *Tango14*$^{KO-WS/+}$, are also rescued. In contrast, R55 does not rescue the dopaminergic neuron and SING defects of group B mutants (Fig. 4c, d, Supplementary Fig. 13c). Interestingly, animals of the subgroup A3 (light blue) including *VAC14*$^{KO-WS/+}$, *VhaM8.9*$^{KO-WS/+}$, *Gdh*$^{KO-WS/+}$ and *eIF4G*$^{KO-WS/+}$ show a rescue of the reduced TH positive area when supplemented with Q10 and some with R55 (for *VhaM8.9*$^{KO-WS/+}$). These genes encode proteins (loosely) involved in "proteostasis/autophagy" and further work is required to identify tools to target this subgroup more specifically. The significant subgroup-dependent rescue by Q10 and R55 (Fisher's exact test, *p* < 0.001) supports that dopaminergic defects originate from cluster-specific pathomechanisms. Hence, our work delineates at least 2 Parkinsonism subgroups of which one is defined primarily by "mitochondrial-defects" (group B) and another primarily by "endo-lysosomal trafficking-defects" and "proteostasis/autophagy defects" (group A).

## Discussion

Using our collection of *Drosophila* Parkinsonism mutants, we classified 24 genes associated with the disease into subgroups; the identity of the genes clustering together suggest each group is characterized by common underlying molecular pathways. These pathways are defined by the known functions of the genes in each group, such as the involvement of *Pink1* and *park* gene products in mitochondrial function control, or the role of Vps35 and iPLA2-VIA in retromer activity[70]. While the functions of some individual gene products have been

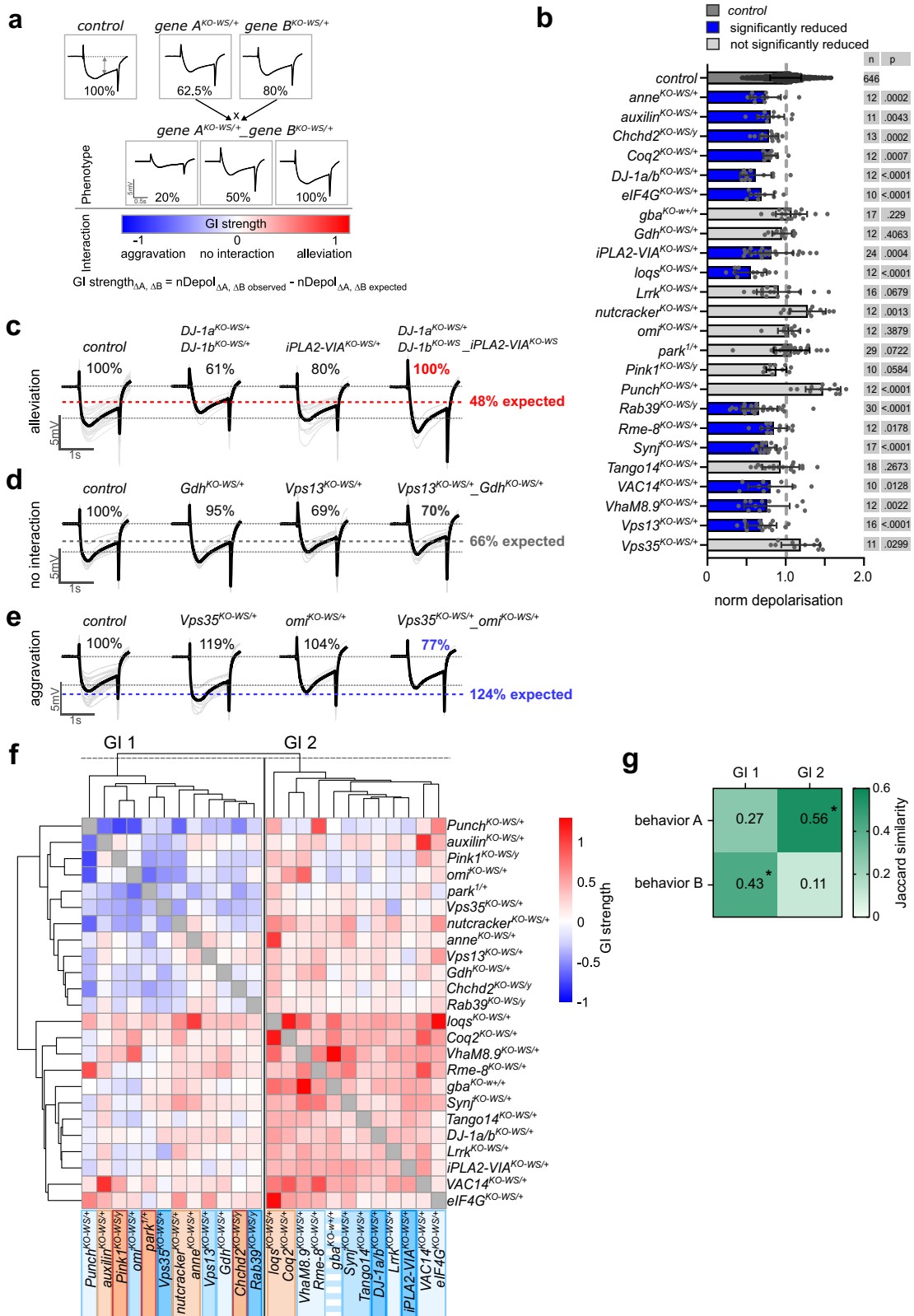

studied before, here we comprehensively map the entire familial genetic landscape of PD and related familial forms of Parkinsonism. Our molecular classification may facilitate the identification of group-specific biomarkers and inform the development of more targeted therapeutic strategies based on underlying disease mechanisms.

The specific causes of most idiopathic PD and related Parkinsonism cases remain unknown. Our analysis thus specifically focused on

familial PD and related genetic causes of Parkinsonism that can be modeled in flies. Although familial Parkinsonism is rare, there is increasing evidence that the pathways involved in familial disease are also relevant to idiopathic cases. For example, mutations in *LRRK2* cause PD and *LRRK2* variants have been associated with PD risk in idiopathic cases. Furthermore, *parkin* loss-of-function mutations cause familial PD, and the protein is often found to be less active in

**Fig. 3 | Genetic interaction screen supports the assignment of Parkinsonism genes to behavior groups.** *Related to Supplementary Data* 1 *and Supplementary Fig.* 12. **a** Scheme of genetic interaction screen using ERGs. The amplitude of the ERG-depolarization of young, single heterozygous mutants (gene A and B) are normalized to the control ($CS^{w1118}w^+$). In the non-interacting model, the expected double heterozygous Parkinsonism mutant effect is computed by multiplying the single mutant effects (depolarization amplitude of 50% in this example) and compared to the observed double heterozygous gene pair. Thereby, the GI strength of gene pairs are defined as "aggravating", "alleviating" or "not interacting". **b** Normalized depolarization of heterozygous single mutants at $4 \pm 1$ d after eclosion. Blue colored bars represent significant decrease compared to control (dark gray), ANOVA Kruskal–Wallis with Benjamini–Hochberg. Bars: mean ± SD; points represent individual animals. *n* and *p* values are indicated in the graph. **c**–**e** Representative ERG traces of alleviating, non-interacting and aggravating genetic interactions: the ERG amplitude of double heterozygous mutants of *iPLA2-*

*VIA* and *DJ-1a/b* is rescued, while the ERG amplitude of gene pairs *Gdh* and *Vps13* show no difference to the expected ERG amplitude and the double heterozygous mutants of *Vps35* and *omi* show an aggravated ERG amplitude compared to the single heterozygous mutants. Black, bold traces show the mean trace, while light gray traces show individual recordings. Dashed lines indicate base line and 100%, respectively, while the bold lines represent the expected ERG amplitudes. **f** Hierarchical clustering on the GI strength of gene pairs identifies two groups GI 1 and 2. Mutants are color-coded according to the behavior groups in Fig. 2e; $N \geq 5$ per gene pair. **g** Heatmap representing the Jaccard similarity index of behavior groups A and B with GI groups 1 and 2. *$p = 0.0124$, compared to random distribution of mutants in GI groups: *p*-values were calculated using a one-sided permutation test (10,000 iterations) based on the Jaccard similarity index. No multiple-comparison correction was applied. Source data are provided as a Source data file. ERG electroretinogram, GI genetic interaction.

idiopathic disease[71,72]. Pathologically, both familial and idiopathic cases often exhibit Lewy body accumulation in the brain, and there is significant clinical overlap between these forms of the disease[73–75]. We therefore surmise that the 24 familial Parkinsonism genes analyzed here capture most of the molecular complexity underlying all PD and related Parkinsonism cases, suggesting that both familial and idiopathic disease can be classified into two broad subgroups, but leaving open the possibility of other groups to exist that are not encompassed by familial disease.

Through detailed behavioral profiling, we cataloged a set of 24 mutants into 2 groups and found that the behavioral features within each group correlate with specific underlying molecular dysfunctions. *Group A* encompasses 3 subgroups with 16 genes. The first subgroups, A1 and A2, encode proteins involved in vesicle trafficking (Synj1, Rab39, Vps13, Tango14), retromer (iPLA2-VIA, Vps35) and lysosome (Gba) function as well as Omi and DJ1. While the function of Omi has not yet been directly connected to vesicle trafficking events, the pleiotropic functions of DJ1 include vesicle recycling during neurotransmission[76,77] and autophagy regulation[78]. Similarly, Rab39 and genes of the third subgroup A3 (*VAC14, VhaM8.9, Lrrk, Rme-8* and *eIF4G*) have previously been associated with autophagic pathway modulation and protein homeostasis[79–87]. The positions of Gdh (encoding Glutamate dehydrogenase) and Punch (involved in dopamine biosynthesis) in this group are currently not explained. Finally, *group B* has 8 genes. Five of these encode proteins regulating mitochondrial function (Park, Pink1, Chchd2, Nutcracker and Coq2). The functions of the other members have not yet been directly linked to these organelles (Auxilin, involved in endocytosis, Loqs, a member of the RISC complex and Anne, the fly homolog of ATP13A2, an endolysosomal transporter). Interestingly, the assignment of genes to these two groups is supported not only by two independent methods of unsupervised data analysis—hierarchical clustering and principal component analysis—but also independently confirmed by an orthogonal experimental approach utilizing genetic interaction studies.

Interestingly, by functionally assessing the electrophysiological neuronal properties of the heterozygous Parkinsonism mutants at a young age using ERGs, we observe significant phenotypes (Fig. 3b). This is in contrast to dopaminergic neuron dependent SING and TH positive area not showing overt differences from controls at this young age (except *gba*) (Fig. 1a and Supplementary Fig. 2a). This could be because of the different genotypes (heterozygous vs homozygous) of some of the animals we used in these assays, but also because different neuronal circuits may be affected differently: visual circuitry already at a younger age affecting the ERG response, and SING circuits only at older age (also consistent with the loss of dopaminergic synapses only visible at older age). Given we did not measure ERGs in aged flies of the exact same genotype as those used for the SING assay,

it is at present not possible to conclude if both assays correlate across our mutants.

This is the first step in a broader effort that will work towards translating these findings to human patients, starting with familial forms of the disease and ultimately also expanding to idiopathic cases. This work, which defined two broad groups, each with individual subgroups, will guide future analyses, focusing on group-specific clinical behaviors (e.g., sleep and olfactory parameters[47,88–90]) combined with biomarker discovery, as in the growing PPMI study[91]. We surmise that identifying clinical behavioral and biological markers will ultimately facilitate the classification of idiopathic cases into one of the appropriate groups. Moreover, by adjusting clinical trial design to these groups, clinical benefits of group-specific treatments could be revealed, previously potentially masked by a broader cohort of patients. For example, the group B-specific effect of Q10 highlights the importance of personalized treatment strategies and could explain the failed phase III Q10 clinical trials in early Parkinson's disease in a non-stratified cohort[92].

Our study identified groups of genes per cluster based on behavioral features using hierarchical clustering as well as PCA and genetic interactions. These three methodologies showed significant correlation, underscoring the robustness of the groups we identified. However, some interesting exceptions underscore the power of our approach. One example is Gba, which—as expected—clusters with vesicle trafficking, retromer, and lysosome factors in group A of the behavior screen and GI group 2. Yet, PCA places it among the mitochondrial genes in group B, indicating that some behavioral parameters of *gba* mutants align with the mitochondrial group. Interestingly, beyond its well-established function in lysosomes[93,94], Gba has recently been found to regulate mitochondrial function[95] and is imported into these organelles to maintain Complex I activity[96]. This suggests that our different analysis methodologies are sensitive to the various aspects of the protein's function.

Our analyses also show a divergent assignment of *Rab39* to subgroups. In the behavioral analysis, *Rab39* grouped with vesicle trafficking, retromer, and lysosome factors in group A. However, in our genetic interaction analyses, *Rab39* clustered with mitochondrial genes in GI group 1. This discrepancy is again highly informative, because *Rab39* mutants are known to cause defects in vesicle trafficking and autophagy[87,97,98], as well as ER stress-induced mitochondrial dysfunction and oxidative stress[99]. These findings suggest that also our behavioral analysis screen and genetic interaction assessment highlight different facets of Rab39's function. For some other genes, explaining their subgroup assignment at this stage may be less evident, potentially pointing to other, yet to be uncovered, functions. For example, *Omi*, that encodes a mitochondrial protease[100–102], is assigned to trafficking group A, and *auxilin*, a co-chaperone involved in endocytosis is grouped with mitochondrial genes in group B[103]. By using

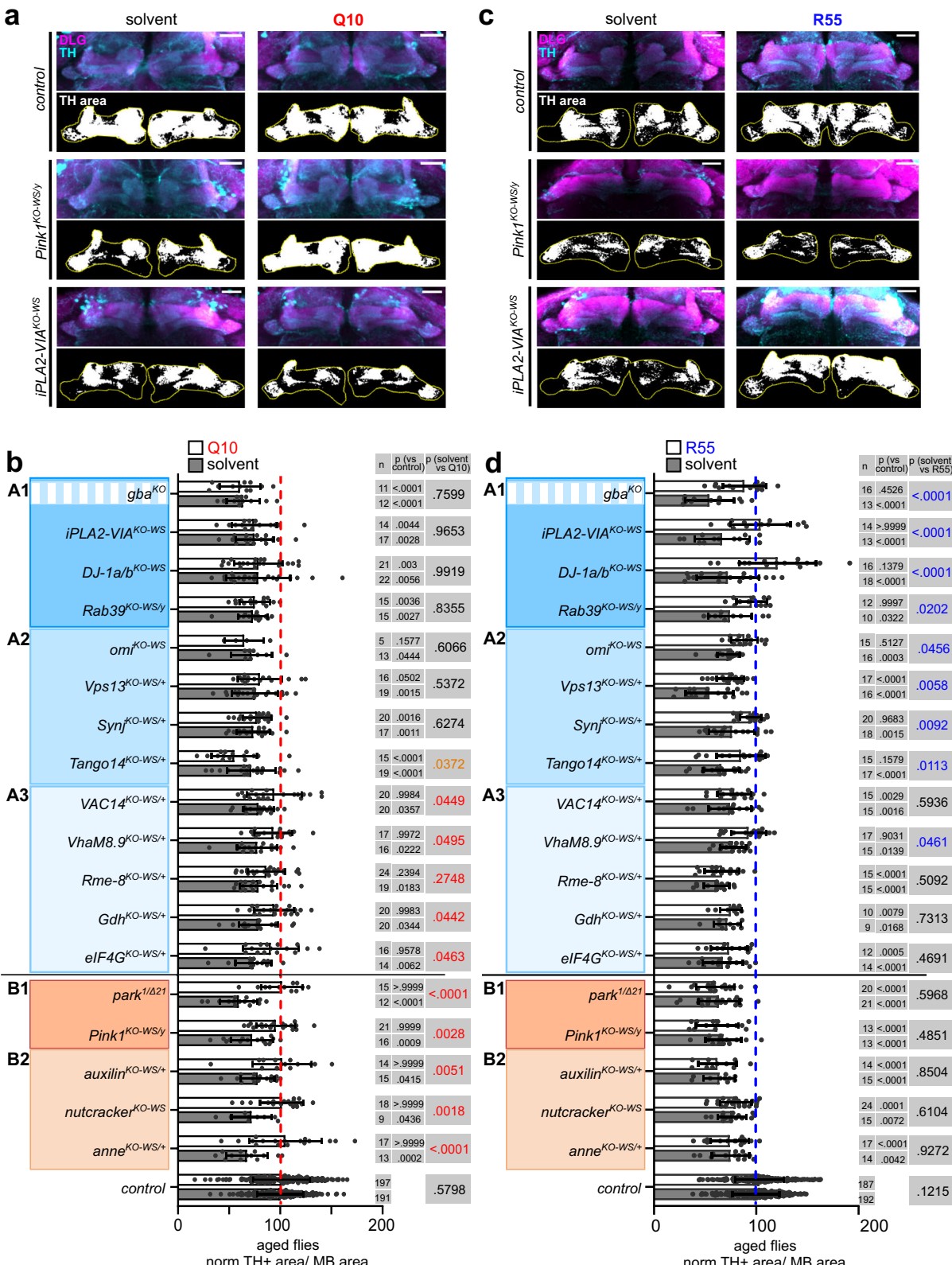

(and in the future also developing) group-specific rescue experiments, like those we included here with R55 and Q10, we can uncover which protein function is the most relevant for maintaining dopaminergic neuron integrity and function.

Our analyses of the genetic space of familial Parkinsonism found two distinct subgroups of genes that are defined by functional molecular pathways and that reveal the complex underlying mechanisms. Future work will expand to include GWAS data and other newly discovered risk factor genes with possibly unknown function, enhancing our understanding of idiopathic disease and ultimately refining biomarker research, stratification in clinical trial design and group-specific drug development.

**Fig. 4 | Dopaminergic neuron defects are rescued by subgroup-specific treatments.** *Related to* Supplementary Fig. 13. **a** Maximum projection confocal image of aged control ($CS^{w1118}w^+$), $Pink1^{KO\text{-}WS/y}$, $iPLA2\text{-}VIA^{KO\text{-}WS}$ fly brains treated with 1 mM Q10 or solvent-control (ethanol) stained with anti-TH (cyan) and anti-DLG (magenta) antibodies, where DLG marks the post-synaptic site of MB. The black and white images represent the thresholded TH area (in white) of "middle z-plane" within the ROI (yellow, outline of MB). Scale bar: 20 μm. Images are a representation of the quantification in (**b**), with n indicated in the graph. **b** Quantification of dopaminergic synaptic area within MB area in aged mutants treated with Q10 (white bars) or control treatment (filled bars) relative to control with solvent treatment. Points are individual animals. Bars: mean ± SD; Two-way ANOVA with Tukey's multiple comparison, *n* and *p* values are indicated in the graph. **c** Maximum projection

confocal image of aged control, $Pink1^{KO\text{-}WS/y}$, $iPLA2\text{-}VIA^{KO\text{-}WS}$ fly brains treated with 12 μM R55 or solvent-control, $H_2O$, labeled with anti-TH (cyan) and anti-DLG (magenta) antibodies and thresholded TH+ area of "middle z-plane" within the ROI (yellow, outline of MB). Scale bar: 20 μm. Images are a representation of the quantification in (**d**), with *n* indicated in the graph. **d** Quantification of dopaminergic synaptic area within MB area in aged mutants treated with R55 (white bars) or control (filled bars) relative to controls with solvent treatment. Points are individual animals. Bars: mean ± SD; Two-way ANOVA with Tukey's multiple comparison, *n* and *p* values are indicated in the graph. **b**, **d** Red and blue *p*-values indicate a significant increase, while orange represents a significant reduction. Source data are provided as a Source data file.

## Methods

### Experimental model and subject details

**Fly stocks and rearing conditions.** The work in this study was overall compliant with institutional ethical standards. Using CRISPR/Cas9-mediated gene editing, we created a comprehensive collection of *Drosophila* loss of function Parkinsonism models. We selected the 24 fly homologs of genes known to be causative or conveying risk for Parkinsonism in humans and inserted an attP-flanked $w^+$ reporter cassette in each of the genes substituting the first common exon of possible transcripts.

The "control" strain denotes a semi-isogenized $w^{1118}$ strain that underwent backcrossing to *Canton-S* for 10 successive generations, resulting in a strain termed *Canton-S-$w^{1118}$*. Additionally, a *Canton-S-$w^{1118}$* strain was augmented with a transgenic mini-white integration on the X-chromosome by microinjection of pFL44S{w + }[104] into the su(Hw)attP8 landing site (BestGene). This modification results in a red-eyed "control" strain named *Canton-S-$w^{1118}$-$w^+$*. The entire genome of *Canton-S-$w^{1118}$* was sequenced, revealing the absence of deleterious mutations in homologs of known causative Parkinson's disease genes, as well as in homologs of human genes proximal to GWAS loci. In order to maintain genetic consistency within the experimental framework, all Parkinson's disease KO lines (see below, except for $park^{1/\Delta21}$) were subjected to over ten rounds of backcrossing with the established *Canton-S-$w^{1118}$* control strain. A thorough verification protocol was implemented for all generated fly lines, which included conducting multiple PCR assays followed by sequencing of the PCR products. Detailed information regarding the primers utilized for this verification process can be found in the supplementary file (Supplementary Document 1). For *park*, the null-mutant fly lines $park^1$ and $park^{\Delta21}$ from the Bloomington *Drosophila* Stock Center were used[26,27].

Flies were maintained on standard corn meal and molasses food. For experiments, flies were raised in parallel on the same batch of food, which was exchanged every 3 to 4 days, in a temperature and light-controlled incubator at 25 °C and 12-h light-dark cycle and kept in mixed populations of similar density. Control and experimental genotypes were aged to 6 ± 1, 26 ± 2 and 42 ± 2 days (d) as indicated for immunohistochemistry and behavior assays except for some Parkinsonism mutants: *nutcracker$^{KO\text{-}WS}$* were tested at 15 d, $Pink1^{KO\text{-}WS/y}$ at 22 ± 2 d, $iPLA2\text{-}VIA^{KO\text{-}WS}$ and $gba^{KO}$ mutants at 25 ± 1 d due to their shorter survival.

For Q10 and R55 treatments, male flies of control and experimental genotypes were collected at 1–3 days post eclosion and aged on treatment food, which was renewed every 2 to 3 days. On the day of the experiment, flies were flipped on fresh treatment food at ZT zero to one. All treatment food was prepared every one to two weeks and kept at 4 °C in darkness. Q10 (Merck) was prepared as a 50 mM stock solution in ethanol (99.8%), aliquoted in single-use portions and stored at −20 °C. R55 (Calbiochem/Millipore) was dissolved in sterile distilled water as a 12 mM stock solution, aliquoted in single-use portions and stored at −20 °C. Q10 and R55 with their respective solvent controls were added to fresh fly food heated up to around 40 °C to a final concentration of 1 mM for Q10 and 12 μM for R55.

Both L-Dopa (Merck) and D-Dopa (Merck) were prepared as a 15.2 mM stock in sterile distilled water on ice and stored at −80 °C in single-use aliquots. Both L-Dopa and D-Dopa as well as an equal volume of the solvent were added to fresh fly food as described above to yield a 1 mM final concentration. Aged male flies were selected and transferred to L-/D-Dopa food 10 days prior to the behavior assay[105].

The fly stocks and genotypes were used and abbreviated in this study are summarized in Supplementary Table 3.

w[1118]; gmRPL11/CyO Tb[1] was a gift from Nicolas Tapon and the *pdf* null mutant, $park^1$ and $park^{\Delta21}$ were obtained from BDSC.

**Generation of Parkinsonism knock-out *Drosophila* models.** Parkinsonism KO *D. melanogaster* strains were established employing the CRISPR/Cas9 methodology. This involved the utilization of both a donor plasmid (pWhite-STAR) and a tandem gRNA expressing plasmid. The donor plasmids were equipped with an Integrase-mediated cassette exchange (IMCE) cassette, which upon integration into the genome, expressed mini-white. Two homology arms facilitating homology-directed repair (HDR) flanked this cassette. HDR is initiated subsequent to the introduction of two double-strand breaks within the DNA encompassing the initial common exon across all possible transcripts. This process culminates in the substitution of the first common exon with the IMCE-cassette. The positioning of homology-directed repair (HDR) was strategically guided by the selection of homology arms, ensuring that the IMCE cassette resided between two non-evolutionarily conserved regions. The donor plasmid also possesses *I-SceI* endonuclease sites. In the rare event that the donor plasmid integrates fully into the fly genome by the recombination of only one homology arm, we can correct the integration by expressing ectopically I-SceI in the fly. This enzyme induces a double-strand DNA break at the *I-SceI* site, initiating a new DNA repair process where the appropriate homology regions can undergo recombination to ultimately yield the correct fly.

**Cloning of tandem gRNA constructs.** Unique guide RNAs (gRNAs) targeting all Parkinsonism genes used in this study were identified through http://crispr.mit.edu/ and subsequently introduced into pCFD4: U6:1-gRNA U6:3-gRNA constructs as per established methodologies outlined in Port et al. (2014)[106] and detailed in http://www.crisprflydesign.org/wp-content/uploads/2014/06/Cloning-with-pCFD4.pdf.

**Cloning of donor plasmids.** Homology arms, spanning 1 kilobase (kB) encompassing the initial shared exon of all gene transcripts, were cloned into either pWhite-STAR[107] or pWhite-STAR2 (see Cloning of pWhite-STAR2) (Supplementary Fig. 1a, b).

Utilizing HiFi DNA Assembly (NEB), four distinct fragments were combined: the mini-white IMCE-cassette, left homology arm, right homology arm, and the plasmid backbone. The homology arms were either synthesized via gBlocks (IDT), which underwent modifications to adhere to IDT specifications, or were amplified through Q5 (NEB) PCR utilizing genomic fly DNA from the target genotype

(*Canton-S-w^1118*). The mini-white IMCE-cassette and vector backbone were derived through restriction digestion of pWhite-STAR using AvrII and XhoI enzymes[107]. To obviate cleavage of the donor plasmid, mutations were introduced to the PAM sequence within the homology arms, predominantly altering NGG to NAA, although alternative modifications were implemented when necessitated, particularly in instances where the PAM sequence resided within coding regions.

To enhance the functionality of pWhite-STAR for backbone integration detection and to prevent the SV40 poly(A)-tail to become functionally silent[108], modifications were made to the original pWhite-STAR construct[107]. Specifically, one *FRT* site and one *I-SceI* site were replaced with a 3xP3-eGFP fluorescent eye marker, enabling the selection against backbone integration. In pWhite-STAR2 the homology arms were cloned such that, upon integration into the genome, the mini-white gene is transcribed in the opposite direction to the Parkinsonism gene. In detail, pWhite-STAR was subjected to XhoI-HindIII digestion, resulting in the excision of one *FRT* and one *I-SceI* site. Subsequently HiFi DNA Assembly (NEB) was conducted using two distinct fragments: a 3XP3-eGFP marker[109] using a gBlock (IDT) and an SV40 poly(A)-tail amplified via PCR from the pUAST.attB plasmid[110] to create the pWhite-STAR2.

For two fly lines, *gba^KO* and *omi^KO-WS*, the strategy had to be adjusted:

The GBA orthologs in *Drosophila* consist of two genes, gba1a and gba1b, located adjacent to each other in the fly genome, with a third gene, CG31413, located between them. To simultaneously knock out both *Gba1a* and *Gba1b*, we engineered homology arms for the donor plasmid such that, after recombination, a stop codon is introduced in *Gba1a*, and CG31413 along with the first exon of *Gba1b* is replaced with our IMCE cassette. In a subsequent step, we reintroduced *CG31413* using PhiC31 integrase-mediated cassette exchange with the plasmid pReC-CG31413 to generate *gba^KO* (*gba1a,gba1b^KO*). The pReC (Rescue Cassette) was generated by linearizing pUC19 with SapI-EcoRI, followed by the insertion of two gBlocks synthesized by IDT. Subsequently, pReC was linearized with XhoI-XbaI and the entire genomic *CG31413* region, amplified via PCR from *Canton-S-w^1118* was cloned into the linearized plasmid trough HiFi DNA Assembly (NEB).

The *omi^KO-WS* stock is homozygous lethal, a phenomenon previously reported by Tain et al. (2009)[111]: The imprecise P-element excision of G4907 (Genexel Inc.) between *omi* and *mRpL11* disrupted not only the genomic region of *omi* but also affected the promoter region and transcription start site of *mRpL11*. Similarly, our strategy also affected *mRpL11* by generating a null allele for *omi*. To address this, we utilized the gmRpL11 rescue construct described by Tain et al. (2009)[111] in our *omi^KO-WS* flies, which successfully rescued the homozygous lethality of our line.

A comprehensive array of primers, oligos, gBlocks, and fly strains utilized is delineated in Supplementary Document 1.

All plasmids were sequence verified, and plasmid injections were performed by BestGene (BestGene Inc., Chino Hills, CA, USA).

**Quantitative RT-PCR.** For each sample, 10 flies: 5 males and 5 females were collected except for *Pink1^KO-WS/y*, for which we only used hemizygous males, at 1–5 days after eclosion. RNA was extracted using Maxwell RSC instrument and Maxwell RSC RNA kit (Promega) and transcribed using the SuperScript VILO cDNA Synthesis Kit (Thermo-Fisher Scientific). Subsequently, cDNAs were quantified by qPCR using a LightCycler 480, 480 SYBR Green master mix (Roche) using qPCR primers enlisted in Supplementary Table 4. qPCR primers were designed using the PrimerQuest Tool (Integrated DNA Technologies). mRNA levels were determined using the Δ-Δ-CT method, where Ct values were first normalized to the housekeeping gene Rp49. Gene expression levels were then plotted as a relative gene expression compared to the endogenous *Drosophila* gene expression in control samples[30].

**Immunohistochemistry and confocal imaging.** Immunohistochemistry of $6 \pm 1$, $26 \pm 2$ and $42 \pm 2$ day old adult brains in at least 2 independent experiments were dissected in ice-cold PBS and fixed for 30 min in freshly prepared 3.7% paraformaldehyde (in 1x PBS, 0.2% Triton X-100 (PBX)) at room temperature (RT). After 3× washes for 15 min in PBX at RT on a shaker, they were incubated in blocking solution (PBX, 10% normal goat serum (NGS)) for 1 h at RT. Subsequently, the brains were incubated in primary antibodies (rabbit α-TH (Millipore Cat# AB152, RRID: AB_390204), 1:200, mouse α-DLG (DSHB Cat# 4F3 anti-discs large, RRID: AB_528203), 1:100) in blocking solution at 4 °C for 1.5–2 days and washed 3× 15 min in PBX at RT. Secondary antibodies (goat α-rabbit IgG Alexa Fluor™ 488 (Life Technologies Cat# A-11034, RRID: AB_2576217), goat α-mouse IgG Alexa Fluor™ 555 (Life Technologies Cat# A-21424, RRID: AB_141780), at 1:500) in PBX with 10% NGS were applied overnight at 4 °C and brains were finally washed 3× 15 min in PBT at RT on a shaker and mounted, anterior facing up in RapiClear 1.47 (SUNJin Lab)[112]. Z-stacks of entire brains were imaged at a Nikon A1R confocal microscope with a 20x (NA0.95) water immersion lens. Acquisition was performed using a galvano scanner, a zoom factor of 1, scan speed 0.5 and line averaging of 2. All images were acquired with a pinhole of 2.3 Airy unit and a resolution of 1024 × 1024. Z-stacks (step intervals of 3 μm) were used in data acquisition and the same image settings were maintained across the genotypes and imaging sessions. Fiji[113] was used for image analysis. Dopaminergic neuron innervation of the mushroom body was quantified as follows: Anti-DLG was used to determine the five z-stacks which include the synaptic region of the mushroom body (MB) lobes. The outline of the MB (region of interest, ROI) was defined in the sum projection of the five z-planes. Within the selected z-stacks, the area of the anti-TH fluorescence was thresholded (default threshold–using the same threshold for all the 5 z-planes) excluding background signal comparable to the control. The thresholded area within the ROI was quantified in every z-plane, summed and normalized to the area of the MB ROI for every brain individually (TH+ area/MB area). For every experiment, the individual TH+ area/MB area values were normalized to the mean of the control. The maximum age of every genotype is plotted in "aged flies". For representative images, the maximum projection of five z-planes as well as the thresholded middle z-plane is shown.

**Startle-induced negative geotaxis (SING).** Negative geotaxis was assessed at Zeitgeber ZT three to five in mixed fly populations or in experiments with treatments (L-/D-Dopa, Q10, R55 and their corresponding controls) in groups of male flies as described previously[114,115]. Flies at indicated ages (Fig. legends) were transferred into the first tube of the apparatus without prior anesthesia and allowed to adjust for some minutes to experimental environment ($24 \pm 1$ °C, 40–60% humidity). They were tapped down five times and allowed to climb up for 30 s, at which time the flies that reached the upper tube were moved to the next tube. The procedure was repeated four times, the number N of male flies with the corresponding genotype were counted per tube to calculate the SING score, which was normalized to the mean SING score of the control[116].

$$SING\,score =$$
$$((N_1 * 0) + (N_2 * 1) + (N_3 * 2) + (N_4 * 3) + (N_5 * 4))/4(N_1 + N_2 + N_3 + N_4 + N_5)$$
$$(1)$$

$N_k$ = number of flies in the $k$th tube.

**Seizures-like behavior assay.** Seizure-like behavior induction was tested in groups of $5 \pm 1$ male flies at $6 \pm 1$ d and $26 \pm 2$ d post eclosion except for *nutcracker^KO-WS* and *Pink1^KO-WS/y*, which were tested at 15 d and at $22 \pm 2$ d, respectively. Flies were transferred into transparent vials, rested for at least 30 min after anesthesia exposure and mechanically stimulated by vortexing the vial for 10 s at maximum intensity[36].

Numbers of resistant flies and flies with seizure-like behavior (involuntary movements or paralysis) were quantified within 10 s post stimulation[117]. Graphs represent the % of flies without seizures. To visualize the convergence of familial Parkinsonism cases reported to have seizures with the Parkinsonism mutants showing seizure-like behavior, we used the R (v3.6.3) and R package VennDiagram (v1.7.3).

**Electroretinograms (ERGs).** ERGs were recorded from young flies $4 \pm 1$ d after eclosion as previously described[63,64]. To limit pigmentation variability due to locus-dependent $w+$ activity in pigment cells of mutant flies, we crossed all control heterozygous flies with $CS^{w1118}w^+$ to achieve $w^+/w^+$ (i.e., wild type red pigment). Moreover, all gene pairs will also have a $w^+/w^+$ (one $w^+$ from each mutant). Flies were immobilized on glass microscope slides, by use of double-sided tape. For recordings, glass electrodes (borosilicate, 1.5 mm outer diameter) filled with 3 M NaCl were placed in the thorax as a reference and on the fly eye for recordings. The flies were exposed to darkness for 3 s, followed by 1 s of LED light illumination. This was repeated 5 times for each fly. Light-evoked signals were amplified by a DC amplifier and the amplified signal was processed by a data acquisition device (Clampex) and Axosope 10.7, connected to a PC running Clampfit 10.7 software (Molecular Devices). ERG traces were analyzed in IGOR Pro 6.37 (WaveMetrics) using a custom-made macro[118].

**Fly behavior monitoring.** The screening of sleep and activity behavior of young male flies was performed using the fly behavior video recording platform ethoscope[119]. Briefly, at 2–5 days post eclosion flies were introduced into glass tubes (65 mm length, 5 mm external and 3 mm internal diameter) with standard corn meal and molasses food (sealed with wax) on one end and closed with cotton wool on the other. In incubators at 25 °C with 12:12 light-dark conditions, 20 flies per ethoscope were recorded for at least 5 consecutive days at 2 frames per second with infrared light and infrared light filters (transmission > 730 nm, LEE Filters)[120]. We screened $n \geq 69$ animals for every genotype in at least 3 independent experiments. The position of each animal was saved at each time point in SQLite files and subsequently analyzed with R (v3.6.3) and rethomics with adjusted R packages behavr, scopr and sleepr[121] (v0.3.99). Using the automatic behavior annotator at a resolution of 10 s, the behavior of individual flies was determined. Thus, in 10 s intervals, flies were scored as moving or asleep. Sleep was defined following the 5-min rule, in which immobility bouts longer than 5 min were counted as sleep bouts, including the first 5 min. A fly was scored as immobile if the distance between 2 consecutive frames in the 10 s interval was less than 0.25 mm (corresponding to a velocity correction factor $a = 0.0042$). Flies that died during the monitoring as well as the first day of recording due to habituation were excluded from further analysis. The following parameters were quantified to assess the activity, the circadian rhythm and sleep architecture of individual flies: Within 5 days, the mean sleep amount per light condition (sleep fraction/day) and per dark condition (sleep fraction/night) were calculated. For the weighted mean latency to longest bout, the time between ZT 12 (= lights off) and the longest sleep bout was determined per night. The weighted mean bout length (mean bout length/night) as well as the weighted mean number of sleep bouts (nr bout/night) were scored during the dark condition. Morning anticipation were calculated as described previously[59]. The mean velocity if awake (velocity if awake) was assessed based on the velocity determined within the 10 s interval annotation 'moving'. The cumulative distance within the 10 s interval was calculated and summed per 24 h to determine the traveled distance (total distance) for each animal per day. Representative sleep traces of genotypes were visualized using the ggetho package in R (v0.3.6).

**Pattern recognition using non-negative matrix factorization (NMF).** The pattern recognition for behavioral analysis was done in three steps. First, the (x,y)-coordinates of each individual fly are transformed into a one-dimensional vector that captures the fly's temporal behavior. Specifically, we counted how often the fly performs a certain type of behavior during each period of the day, aggregated over the period of the experiment. Second, the vectors of all flies are concatenated to form a matrix and non-negative matrix factorization (NMF)[51,122] is applied to automatically cluster common behavioral patterns. This method effectively identified shared features in the behavior of (a subset of) all flies. Ultimately, the behavior of individual flies can be described by a weighted combination of these shared features (e.g., waking up early, more activity during daylight periods). The weights are used to compare the behaviors across different flies, enabling the identification of groups of flies exhibiting similar behaviors. The implementation of the NMF method is provided by Scikit-Learn[122] version 1.0.1 for Python 3.9.6.

**Constructing vectors of temporal behavior.** For each fly $f$, a vector is created that summarizes during which periods of the day the fly performs a certain type of basic behavior. Therefore, we first categorized the behavior of a fly in each individual frame based on the fly's corrected velocity, as discussed in Geissmann et al. (2017)[119]. For each frame $i$ sampled at timestamp $t$, the corrected velocity $v$ is computed as:

$$v_i = \frac{10^{\frac{d_i}{1000}}}{(t_i - t_{i-1}) * FPS * a} = \frac{10^{\frac{d_i}{1000}}}{a} \qquad (2)$$

where $d$ is the log-transformed Euclidean distance between successive frames. The velocity correction factor $a = 0.0042$ was empirically defined.

Second, we created bins for each minute of the day (24 h × 60 min = 1440 bins) and aggregated the behavior of each fly in each bin over the duration of the experiment (5 days and nights; excluding the first day of recording due to habituation) obtaining a vector $x_f \in \mathbb{R}_+^{1440}$. Similar to the rethomics-based analysis we used the aggregation approach based on "Time Spent Sleeping", in which the percentage of frames where the basic behavior was "immobile" for 99% of the frames in the preceding 5-min period is calculated. To promote smoothness in the values of successive 1-min bins, a Gaussian blur[123] with $\sigma = 5$ is applied to the vector $x_f$. This process replaced the value of each bin in $x_f$ with a weighted average of itself and its neighboring bins, resulting in the blurred vector $x_f' \in \mathbb{R}_+^{1440}$.

**Identifying behavioral patterns.** For each fly and behavior, we now have a single one-dimensional vector $x_f' \in \mathbb{R}_+^{1440}$ that summarizes the behavior of the fly in function of the time of the day. We then construct the matrix $M_b = [x_0 x_1 ... x_l]$ that contains as columns the vectors of all $l$ flies in our dataset for behavior type $b$. Next, we compress matrix $M$ by applying NMF, which is a form of principal component analysis where the resulting components contain numbers ≥0. This results in two matrices $W$ and $H$ such that:

$$M \approx WH, \qquad (3)$$

where $M \in \mathbb{R}_+^{1440 \times l}$, $W \in \mathbb{R}_+^{1440 \times k}$, and $H \in \mathbb{R}_+^{k \times l}$. Here, $k$ is a user-defined parameter that refers to the number of principal components for behavior type $b$.

The columns of $W$ are the principal components that represent basic temporal patterns of behavior. The rows of $H$ are compressed versions of the vectors in $M$. In other words, if the vector $x$ was the $i$-th column in matrix $M$, then the $i$-th row of $H$ is its compressed vector. By multiplying the compressed vector with the principal component matrix $W$, a vector similar to the original can be reconstructed from only $k$ features.

To determine the optimal value for $k$, we used the elbow method[53]. This standard approach plots the reconstruction error (explained variance score) between $W \times H$ and the original matrix $M$ over different values of k and looks for an "elbow" (i.e., inflection point), which revealed 5 meaningful and distinct patterns in this case (Supplementary Fig. 4e). Additionally, we explored the use of NMF for clustering Parkinsonism mutants based on their dominant behavioral pattern. For this, we assigned each mutant to the cluster corresponding to the maximum component in its encoding vector (i.e., cluster(i) = argmax$_j$W$_{ij}$), then computed the Calinski–Harabasz Index[124] across different k for the obtained clustering. The score was maximized at k = 2, indicating that two clusters optimally separate the mutants, but k = 5 offered a reasonable balance between more fine-grained clusters and cluster quality (Supplementary Fig. 4f).

**Analyzing behavior as a combination of patterns.** Each value in a fly's compressed vector is interpretable in the sense that its numeric value quantifies how often the fly executes behaviors of type $b$ at periods of the day in the temporal group of a specific principal component. Thus, we quantified how strong each pattern is reflected in the behavior of each fly by aggregating the distribution of principal component values for every fly within each genotype group (Supplementary Fig. 4g–k).

**Correlation, cluster and principal component analysis of behavior screen.** Correlation analysis for the SING score and TH+ area/MB area as well as velocity if awake vs sleep features and NMF components were computed based on the mean values of every genotype in GraphPad Prism 10.1.2 (San Diego, USA). Linear regression plots show the 95% confidence intervals (gray area) and $R^2$ and $p$-values.

For the cluster analysis, the morning anticipation was excluded, as it would be highly correlative with the NMF components. The means of the activity and sleep parameters as well as NMF components were calculated for each genotype, scaled and hierarchically clustered based on correlation distance and mcquitty-linkage with the R pheatmap package (v1.0.12). Similarly, the dataset was subjected to dimension reduction principal component analysis using the prcomp function and R ggbiplot package (v0.6.2). The scree plot describing the proportion of variance explained by each principal component was determined by dividing the squared standard deviations of the principal components by the total variance and the cor2-values were obtained with R corrplot package (v0.92).

**Parkinsonism gene expression clustering across cell-types of the fly brain and body.** To assess whether Parkinsonism gene expression differences across the fly brain might explain the different sleep and locomotion phenotypes, we re-mined our previous single-cell RNAseq atlas[125]. We selected only control cells at 5 days of age, since we were interested in potential baseline differences at a young time point, where also our behavioral analysis was carried out. To perform hierarchical clustering, we first normalized the counts and took the natural logarithm +1, followed by taking the average for each cell-type. After scaling we then used the scipy linkage and dendrogram functions with the wpgma method and correlation distance. To analyze PD expression across cell-types of the fly body, the Fly Cell Atlas "body" dataset was downloaded from https://flycellatlas.org/#data[126]. The log1p normalized counts of each cell-type were first averaged and then clustered as described above.

**Genetic interaction analysis.** In a trans-heterozygous screen, the amplitude of the ERG depolarisation of every heterozygous Parkinsonism mutant (except for Pink1$^{KO-WS/y}$, Chchd2$^{KO-WS/y}$ and Rab39$^{KO-WS/y}$ we resorted to hemizygous flies, as their genes are located on the X chromosome) as well as of all 276 gene pairs were recorded and normalized to the control (CS$^{w1118}$w$^+$) of the individual experiment. To compute the expected depolarization amplitude for every gene pair in

a non-interacting model, we multiplied the normalized mean depolarization amplitude of the single mutants[65,66]. Next, the genetic interaction (GI) strength was calculated by subtracting the modeled expected depolarization from the individual observed depolarization amplitude of double heterozygous gene pairs. If the expected and the observed amplitudes are the same (GI strength = 0), the genes are not genetically interacting. Other gene pairs exhibit an "aggravating" or "alleviating" genetic interaction when expected and observed values diverge from another (Fig. 3a). Following prior genetic interaction studies we consider a genetic interaction only when |GI strength| ≥ 0.3. Gene pairs with scores within −0.3 to +0.3 were treated as no/weak interaction[127,128]. The GI profiles of Parkinsonism mutants were hierarchically clustered based on Euclidean distance and mcquitty-linkage with the R pheatmap package (v1.0.12).

**Bayesian analysis.** Because the normalized mean depolarization amplitude of the single mutants ignores the variance observed in biological replicates, we opted to additionally run a Bayesian analysis approach aiming to quantify the uncertainty of the results by modeling the consistency of the biological replicates. We defined the following random variables to build our Bayesian model:

– α observed depolarization of *the first single mutant*, relative to the mean of the controls of the same day.
– β observed depolarization of *the second single mutant*, relative to the mean of the controls of the same day.
– ɩ value which models deviations from the expected value under the no-interaction assumption as an additive amount (so deviations from 0 indicate an interaction).

We built our model so that:

$$\alpha \sim Beta(2, 5)^*1.05 \qquad (4)$$

$$\beta \sim Beta(2, 5)^*1.05 \qquad (5)$$

$$\daleth \sim Normal(0, 1) \qquad (6)$$

We decided to model α and β following a Beta distribution with parameters 2 and 5 because it has the following characteristics:
– A posterior probability distribution between 0 and 1 (because they represent fractions of the control). The multiplication by 1.05 makes it possible to be slightly above 1, to model cases where minor increases may be possible.
– Asymmetry, with a shift to values higher than 0.5, to capture fractions closer to 1.

We decided to model ɩ following a Normal distribution centered at 0, because we wanted to impose very little prior knowledge about whether the gene pair interacts, and with a standard deviation of 1 to model noise in the readings.

This way, the contribution to the observed depolarization by each genotype, normalized with the WT control is as follows:
- Single mutant 1 = α
- Single mutant 2 = β
- Double mutant = α * β + ɩ.

Techniques like Markov Chain Monte Carlo (MCMC) sampling allow us to estimate the posterior probability distribution for each of these variables (i.e., an estimate how likely every possible value is that these variables can take), given the observed data. This way, if 0 is not within the area with the most probability for ɩ, the model is telling us the observed data is consistent with some sort of interaction between the genes. Our definition of the area with most probability is the 95%

High Density Interval (HDI), i.e., the smallest interval of possible values of ι, which contains 95% of the area under the posterior distribution.

The implementation of the model and the MCMC sampling is provided by the PyMC package version 5.10.4 for Python 3.10.14.

**Jaccard similarity and random distribution analysis.** To describe the similarity of behavior-defined groups and genetic interaction groups as well as the stability of behavior-defined groups in the absence of one behavioral feature at a time, we used the Jaccard similarity function, which computes the intersection over union (IoU) for any two groups[51]. The similarity for each combination was calculated and visualized in GraphPad Prism 10.1.2 (San Diego, USA).

The stability of clusters was determined and expressed in [%] by counting the Jaccard similarity index for every subgroup (A1–B2) in categories of 0–0.34, 0.35–0.69, and 0.7–1.

To estimate the significance of the observed similarity, we generated 10,000 simulated sets of random groups of genes with the same size distribution as the genetic interaction groups, and computed how frequently a behavior-defined group had a higher similarity with one of these groups than with a real genetic interaction group. The fraction of simulations (out of 10,000) where the similarity is higher in the random group constitutes our $p$-value (the expected probability of a more extreme similarity under the null-hypothesis). The code for this analysis was implemented in R 4.2.3.

**Quantification and statistical analysis.** GraphPad Prism 10.1.2 (San Diego, USA) was used to determine statistical significance. Datasets were tested for normal distribution using the D'Agostino-Person Omnibus and Shapiro–Wilk normality tests. For non-normally distributed datasets ANOVA Kruskal–Wallis test followed by a Benjamini–Hochberg post hoc test (q = 0.05) was used for multiple datasets. When multiple parameters were compared (genotypes and treatments) a two-way ANOVA was used, followed by a post hoc Tukey test for multiple comparison correction. Significance levels are defined as $***p < 0.0001$, $**p < 0.01$, $*p < 0.05$ and ns, not significant. '$n$' in the legends indicates the number of animals used and analyzed. In the SING and seizure assay, '$n$' indicates the number of times the experiment was performed with groups of animals. The association between genotype subgroups (A1–B2) and small molecule treatment (Q10 vs R55) was tested in R (version 3.6.3). A contingency table was generated, and significance was assessed with Fisher's exact test, appropriate for small sample sizes. Data are plotted as mean ± SD or median and IQR. Specifics on the statistical test used for analysis are reported in the figure legends.

### Reporting summary
Further information on research design is available in the Nature Portfolio Reporting Summary linked to this article.

## Data availability
The data generated in this study are provided in the Source data file with this paper. Further data are available upon request. Source data are provided with this paper.

## Code availability
The code for data analysis can be found at https://github.com/verstrekenlab/drosophila-Parkinsonism-subgroups[129].

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

## Acknowledgements

We thank the VIB BioImaging core and the Single Cell Bioinformatics Unit (Kristofer Davie), the Vlaams Supercomputer Centrum, VIB BITS (Janick Mathys), Prof. Dr. Wim Vandenberghe and the members of the Verstreken lab for discussions. Research support was provided in part by ERC (101054310 – P.V.), the Chan Zuckerberg Initiative (DAF 2018 - 191857 (5022) P.V.), a Methusalem grant of the Flemish government (METH/21/05 BDS JdW P.V.), FWO Vlaanderen (The Role of Parkinsonism Genes in Synaptic Autophagy, G031324N P.V.; Synaptic Autophagy: A Common Culprit in Parkinson's Disease, G0B8119N P.V.; Olfactory defects in Parkinson's disease at single-cell resolution, G0A5219N P.V.), Aligning Science Across Parkinson's (ASAP-000430 P.V.) through the Michael J. Fox Foundation for Parkinson's Research (MJFF), Opening the Future (ENA-OPTFU3A-O2010 BDS P.V.), the Fund Jacqueline Cigrang managed by the King Baudouin Foundation to P.V. and the KU Leuven Parkinson Fonds to P.V. as well as by VIB to P.V. and S.L. J.D. received support from the Flemish Government under the "Onderzoeksprogramma Artificiële Intelligentie (AI) Vlaanderen" program. N.K. and R.P. were supported by an EMBO long-term postdoctoral fellowship (ALTF 299-2019 N.K., ALTF 980-2019 R.P.), E.N. was supported by postdoctoral fellowship from the FWO Vlaanderen (1282123N), A.K. was supported by a PhD fellowship from the FWO Vlaanderen (11E2223N) and U.P. was supported by the DFG (PE2759/1-1). P.V. is an alumnus of the FENS-Kavli Network of Excellence. All scientific illustrations were created using BioRender.com under an academic license.

## Author contributions

N.K., S.V., and P.V. conceived and designed the study. N.K., J.S.V., P.R., R.P., A.O., N.S., S.V., E.-S.B., J.D., S.L., and P.V. developed the methodology. Software was developed by N.K., J.S.V., P.R., R.P., and A.O. Validation was performed by N.K. and P.R. Formal analysis was conducted by N.K., P.R., R.P., A.O., and L.G. Investigation was carried out by N.K., J.S.V., P.R., R.P., A.O., E.N., L.G., A.K., D.C., N.S., U.P., S.K., S.V., E.-S.B., and J.S. Resources were provided by P.V., S.L., and J.D. Data curation was performed by N.K., N.S., and S.K. The original draft was written by N.K. and P.V., and all authors contributed to reviewing and editing the manuscript. Visualization was performed by N.K., P.R., R.P., and N.S. The study was supervised by P.V., J.D., and S.L. Project administration was carried out by N.K. Funding was acquired by P.V., N.K., R.P., E.N., A.K., U.P., J.D., and S.L.

## Competing interests

P.V. is the scientific founder of Jay Therapeutics. J.S.V. is an employee at Muna Therapeutics. All other authors declare no competing interests.
