## [Transparent Peer Review file · Nature Communications]

Behavioral screening defines the molecular Parkinsonism subgroups in *Drosophila*

Corresponding Author: Professor Patrik Verstreken

Version 0:

Reviewer comments:

Reviewer #1

(Remarks to the Author)

Kaempf et al. have performed a targeted screen using *Drosophila* to systematically study several dozen genes implicated in PD and other genetic/neurodegenerative disorders that cause parkinsonism. They characterize locomotor impairment, recovery from vortexing, dopaminergic neuron loss, sleep, and electroretinograms. They classify genes into several groups based on the pattern of phenotype observed. They also interrogate pairwise genetic interactions to define potential pathways, and test drugs predicted to modulate mitochondrial energetics and vesicular trafficking. Overall, this work provides a valuable compendium of reagents and data for further cross-species genetic dissection of familial forms of parkinsonism, and begins to define genetic subgroups sharing common molecular mechanisms.

Please consider the following suggestions for improving the manuscript:

-I believe that the term “parkinsonism” needs additional qualification for use throughout this manuscript. “Parkinsonism” refers to a rather non-specific human motor syndrome—a pattern that usually localizes to basal ganglia dysfunction. This is similar to other human neurologic motor syndromes, such as “spastic paraparesis” or “ataxia”, which usually result from upper motor neuron or cerebellar dysfunction, respectively. Thus, parkinsonism has anatomic localizing value but does not imply a specific molecular etiology. Drugs / toxins, vascular lesions (stroke, bleeds), and even brain tumors can also cause parkinsonism. This manuscript is focused on genetic/neurodegenerative causes of parkinsonism rather than “parkinsonism” more generally. This is where the *Drosophila* model really is most helpful due to genomic conservation to examine consequences of manipulation of gene homologs for dopaminergic neuronal health. By contrast, there are no basal ganglia in the invertebrate nervous system—making it a poor model to examine higher-level circuit level contributions to parkinsonism. I would therefore recommend that this be better explained in the introduction. Perhaps clarify that the most immediate interest is in “Parkinson’s disease and related genetic/familial causes of parkinsonism”. The discussion of non-motor manifestations (Lines 56-57) is also confusing, and perhaps a distraction—consider deleting. These features are not part of “parkinsonism”, nor do they localize to dopaminergic neuronal / basal ganglia dysfunction, but may be seen accompanying parkinsonism in PD. The term is less relevant in thinking about other non-PD forms of parkinsonism. In the discussion, “the specific causes of most idiopathic parkinsonism cases remain unknown” (line 397)—Are the authors intending here to mean idiopathic PD? This would make the most sense. Similarly, I recommend revising the text later in this paragraph (lines 405-409): “...capture most of the molecular complexity underlying all parkinsonism cases”. For clarity, it would best to specify “PD and related genetic causes of parkinsonism”, rather than other causes of parkinsonism, such as cerebrovascular pathology.

-Very limited information is provided on how genes were selected? Were there specific search criteria or was this based on certain review articles? The diseases under consideration are rather heterogeneous, including both pediatric and adult conditions. Some are considered primary dystonias, in which parkinsonism is a secondary or a more variable feature. Given the importance of gene selection to the design of the study, I would consider including a descriptive table in the main text, perhaps a condensed version of Supp Table 1. I think it is also important to mention what (if any) human genes were excluded due to poor conservation in *Drosophila*. It is useful to classify the different diseases being considered under the umbrella of genetic causes of parkinsonism, such as Parkinson’s disease (GBA, LRRK2), autosomal recessive juvenile parkinsonism, dystonia-parkinsonism, NBIA, etc. I’m not certain that the distinction of typical vs. atypical in Supp. Table 1 is particularly useful—I would consider most of the genes/disorders being included here as “atypical” causes of parkinsonism, with PD being the most common cause of neurodegenerative parkinsonism, and therefore “typical” (e.g., GBA, LRRK2).

Some of the genes included, such as COQ2, NUS1, eiF4G1 and HTRA2/Omi are controversial with limited replication or non-replication. It may be reasonable to include them as part of the screen, but it would be best to note the caveats. If these are indeed not true genetic causes of parkinsonism, it could skew / bias the clustering.

-Related to the above, I believe it is incorrect to talk about a fly recapitulating “hallmarks of parkinsonism”, since flies do not have basal ganglia, and except for movement speed, other essential features that define parkinsonism (rigidity, tremor, postural reflexes) are of course not assessed. Perhaps note instead that flies recapitulate “dopamine responsive, progressive motor impairment”.

-Based on Fig 1a, it appears that different genes are being tested in heterozygosity versus homozygosity (or hemizyosity), with most being tested as heterozygotes? What is the rationale / justification? Was this based on inheritance model? This does not seem to be the case. For example, ATP13A2 causes recessive parkinsonism, but the fly mutant, anne, is tested in heterozygosity? GBA on the hand usually is associated with PD in heterozygosity, but it appears that homozygous KO is tested. What about embryonic or adult lethality?—this information does not appear to be provided anywhere.

-lines 160-171: seizure is not a common accompaniment of PD and neurodegenerative parkinsonism. Moreover, “seizure-like behavior” following vortexing of flies is not necessarily representative of the pathophysiology underlying an epileptic seizure—caveats should be noted. Why is this considered a “seizure” assay, rather than recovery from a kind of trauma—head trauma is also a cause of parkinsonism...

-lines 272-283: It is possible that genes participate in common pathways, yet may not be expressed in the same cells but rather in the same multicellular functional circuits comprised of distinct cell types?? Perhaps consider this as a discussion point.

-Fig 3b: why was ERG rather than locomotor testing used for the gene-gene interaction analysis? Some additional justification/rationale is needed here? A preponderance of genes have ERG deficits in heterozygosity—is this result expected? It would be interesting to test whether any of the interactions are consistent using the SING assay.

-lines 317-319: I would recommend softening the claim that behavior and gene interaction analysis using ERG “define the same 3 parkinsonism subgroups”. While modest overlap was found, clearly the classifications also diverge in important ways.

-340-358: Were additional genes tested with drug feedings for rescue within each group (A, C) besides the examples shown?

-As the authors are aware, the main pathways / gene clusters, such as mitochondrial quality control and endocytic trafficking are well established, and this provides an opportunity to show how the method applied re-demonstrates known relationships among genes. Thus, the classification of genetic causes of parkinsonism into groups would be most compelling if it immediately allowed novel predictions that could be validated from human data. While it may not be reasonable to include this in the current manuscript, maybe the authors speculate on what type of prediction might be made for consideration in future analyses? Could this guide gene discovery or therapy? One potential future approach to consider in the discussion is to apply the method to candidates from PD GWAS, where dozens of potential genes might be assigned to the clusters defined from Mendelian forms of PD / parkinsonism.

Some addition minor questions / suggestions:

-Lines 111-113: the description gives the impression that the SING phenotype may be specific for DA neuronal defects; however, many other CNS (or PNS) lesions can cause SING phenotypes.

-Fig 1: rather than “young” and “aged”, it would be best to note the exact age ranges in the fig.

-lines 155-157: perhaps note that only “selected mutants” were tested with H2O controls (Supp Fig 1g)

-lines 184-185: it may be helpful to many readers to include additional discussion of the sleep disruptions that occur in PD/parkinsonism and what is know about anatomy / pathophysiology to put the screening based on this phenotype into a bit better context.

-line 352: “We challenged this result”—unclear what this phrase means?

-lines 359-362: these final sentences seem better suited to discussion. More context and citations are suggested for the claim that group B is related to autophagy/proteostasis.

Reviewer #2

(Remarks to the Author)

The manuscript by Kaempf, et al. is extremely impressive and exceptionally significant. The authors generate comparable knockout mutations in 24 genes in Drosophila that are orthologous to genes strongly associated with Parkinson Disease in

humans. They then performed unbiased assays investigating a variety of behavioral phenotypes of these mutant flies and using them to cluster the mutations into groups, as well as systematically performing a genetic interaction test among the mutations based on neuronal function. Quite remarkably, the gene clusters suggested by the multidimensional behavioral analysis show very good concordance with the pattern of genetic interactions, suggesting that the two orthogonal approaches are clustering PD risk genes consistently into the same three discrete classes. Finally, a pharmacological test provides an initial suggestion that the classes defined by phenotype and genetic interaction may also serve to stratify individual mutations into classes that are differentially sensitive to particular drug treatments.

A large number and variety of genetic loci have been linked to PD, as is also true of other neurodegenerative diseases. Like those other diseases, while the genetics have highlighted a variety of molecular pathways as potentially being implicated in PD it has been exceedingly difficult to see an underlying order in that diversity of genetic causes and predisposing factors in PD. The current work potentially provides a unique insight into the genetic structure of the disease by giving unbiased, systems-based grounds for pulling together different genetic loci into a small, defined collection of experimentally-motivated groupings. In addition to the value of the specific results, it provides a template that can and should be used with other datasets to test and expand the analysis here.

I have one more significant comment and one or two minor comments

1. I was disappointed that results of the pharmacological test (Fig 4) were reported for only 4 of the 24 mutations. The concordance of treatment response with gene groupings is certainly suggestive, but the number of genes tested is sufficiently limited to make one worry that this is just happenstance. If possible, it would certainly be preferable to see the results of these pharmacological tests across the whole collection of mutants for at least one of the two assays of Fig 4, if not for both. One doesn't expect to see perfect agreement, just as the concordance between the behavioral and ERG assays is not perfect and is not expected to be, but as it stands it is not clear how strongly to interpret this result.
2. Regarding the ERG assay, it sounds like the flies were w- at the genomic locus, with w+ introduced in the knock-in procedure (and therefore with different amounts of w+ activity in pigment cells in different lines, depending on the locus). Is that correct? At the age the assay is done, is the w+/- status of the fly expected to affect the ERG results? It would be worthwhile to address this explicitly in the text for the sake of readers who are not familiar with the assay.
3. This is a very minor point, but my download of the supplements had a blank page for Suppl Table 2, the values of the genetic interaction screen.

Reviewer #3

(Remarks to the Author)

The authors generated a new collection of fruit fly mutants for 24 PD-related genes and used it to group these genes into three subgroups with similar functions and behavioral activities. Since *Drosophila* has been used very efficiently in previous studies to understand the pathological functions of several PD-related genes, this study can be a good starting point for understanding the molecular mechanisms of all currently known PD-related genes whose molecular mechanisms have not yet been clearly elucidated.

Major Points:

1. It is commonly accepted that disrupting the first exon of a gene typically leads to the downregulation of its expression. However, as the techniques the authors employed were novel and these flies were firstly reported in this study, the authors should provide data on the target gene expression in homozygous and heterozygous mutants to confirm that they are indeed true genetic knockouts.
2. Measuring TH+ area relative to the mushroom body area is a very interesting and effective approach. Additionally, the observation that this index is primarily age-dependent and correlates with motor activity is intriguing. However, it is important to note that the main function of mushroom body in fruit fly is not closely related to the main symptoms of Parkinson's disease (parkinsonism). Clarifying this point would help readers better understand the relevance of this new readout.
3. The authors compared parkinsonism phenotypes in Figure 1 using only genetic knockout flies, which is a logical approach. However, in subsequent figures, such as Supplementary Figure 2, where seizure phenotypes are addressed, the authors clustered PD genes using data from both humans and flies. Therefore, it is crucial to consider whether the mutations in PD genes result in loss-of-function or gain-of-function effects. For example, since pathogenic mutations in LRRK2 are known to be gain-of-function, it would be more appropriate to use LRRK2 gain-of-function mutants, such as G2019S mutant, for a more accurate analysis.
4. The connection between the ERG/sleep phenotypes and MB TH+ area is ambiguous. It would be helpful to include correlation graphs, similar to the correlation graph between SING and TH+ area in Figure 1e, showing the relationship between ERG/sleep and MB TH+ area.
5. This study further elucidates the genetic interactions among genes within the same subgroup. However, while the authors used chemicals such as Q10 and R55, it is important to examine whether other known genes within the three clusters have functional interactions with the PD-related genes discussed in this manuscript. For example, it would be crucial to demonstrate whether vesicle trafficking-related PD genes, such as Synj1, Rab39, Vps13, or Vps35, functionally interact with other known vesicle trafficking genes. To address this, the authors should consider selecting genes known to be involved in vesicle trafficking, such as sec22, and performing overexpression or knockdown experiments in a Synj1 (or other vesicle

trafficking-related PD genes) knockout background.

Reviewer #4

(Remarks to the Author)

In this manuscript, Kaempf and co-workers develop a new library of 24 *Drosophila* models of familial parkinsonism using CRISPR/Cas mediated knockout of homologous genes known to cause parkinsonism in humans. The authors carefully backcrossed their mutant lines for >10 generations to the same control isogenic stock. These genetically well-controlled lines are likely to be a great resource to other related studies. The authors identified three clusters of mutants that overlap in mitochondrial function, retromer/vesicle trafficking and proteostasis/autophagy. The construction of behavior-based clusters with links to common biology of a complex disease is particularly attractive, as it greatly raises the chance that the results can potentially accelerate biomarker discovery and drug development. The results show significant reduction of climbing performance with age as well as reduction of TH positive area for most of the mutants screened. Climbing performance was ameliorated when feeding L-Dopa. Climbing and TH+ area was partially restored after feeding with subgroup specific treatments for mitochondrial dysfunction and endo-lysosomal trafficking.

The manuscript is well-written and sufficiently rigorous in most places. The studies are conducted with large numbers of animals and the methods are fairly detailed.

A few points of concern are outlined below:

Major concerns:

1. Missing alpha-synuclein staining data: The authors screened the mutants using climbing to measure motor performance and immunostaining with anti-TH (dopaminergic neurons) focusing on MB. Was there alpha-synuclein accumulation? Since these genes have been linked with other neurological disorders, including alpha-synuclein staining would be important for providing more direct conclusion as to these models recapitulating PD specific pathology.

2. Robustness of the hierarchical classification of mutant sleep patterns: It is not clear why it was necessary to combine NMF features with user-defined behavioral features. How many clusters emerge if only NMF (more unbiased of the two) is used? In using NMF, the use of the "elbow method" is somewhat risky to this referee as this method does not have a rigorous statistical foundation and is widely considered heuristic, at best. The authors may want to consider better alternatives such as those developed by Tibshirani, Walther & Hastie (2001) and/or Calinski and Harabasz (1974). A related question: how sensitive is the clustering to the choice of the number of total features? Lastly, how do we know if the user-defined features are independent of the NMF features since they are used together?

Did the authors examine locomotor data instead of sleep? Locomotor data may contain more movement details that are "averaged out" in the process of constructing sleep timeseries.

3. Use of ERG to characterize parkinsonism: It is unclear what is(are) the rationale behind choosing ERGs as a readout for mutants that showed primarily dopaminergic impairment (Figure 3). The major neurotransmitter involved in *Drosophila* phototransduction is histamine.

The ERG signal is a summation of sustained receptor potentials (depolarization) in photoreceptors and the activity of downstream laminar neurons that are receiving synaptic input from the photoreceptors. Therefore, why was only depolarization change measured and ON and OFF transients excluded? Moreover, due to other factors such as pigment cells that affect the initial depolarization response after light stimulation, measurement of receptor potential (depolarization) is commonly measured when it reaches plateau [Belusic, G., ERG in *Drosophila*, in *Electroretinograms*. 2011, InTechOpen; Vilinsky I, Johnson KG. *Electroretinograms in Drosophila: a robust and genetically accessible electrophysiological system for the undergraduate laboratory*. J Undergrad Neurosci Educ. 2012].

In addition, why was 4 DAE chosen for ERG readout when other experiments were performed in aged flies given the evidence for normal synaptic innervation in young flies (Figure 1)? The traces shown in Figure 3 do not show significant synaptic dysfunction. The authors conclude that some combinations are alleviating, but this measurement is not tracked with age.

Moreover, Figure 1 results show significant reduction of climbing performance with age as well as reduction of TH positive area for most of the mutants screened. This data is not consistent with the reduction of depolarization from ERGs in Figure 3. Some mutants such as nutcracker showed significant reduction of TH+ staining and climbing performance but no change in depolarization. As well as others showed no significant reduction in TH positivity or climbing, but significant depolarization. These discrepancies should be discussed in more detail.

4. Scaled "velocity" in Method details Constructing vectors of temporal behavior: If $d = \log(\text{pixel distance})$, using $v = 10^{(d/1000)}/0.0042$, I get only values of $v > 240$, since $1/0.0042$ is the minimum. Is that correct or is there a typo or did I misunderstand? How was this expression for v derived and why was it necessary to re-scale movements? Either way, the 3 speed categories are arbitrary. It is not clear why the authors use rescaled value of velocity (actually speed, if I understand correctly, since velocity is a vector) in Supplementary figure 4 and not the values determined in mm/sec directly from the videos.

Additional (less critical) comments:

1. Fig 1. Panel F second p value seems to have wrong > direction (VAC14).
2. Fig. 2. How do non-parkinsonism flies, such as pdfKO, cluster with the mutants considered in this study?
3. In Fig 3c, what are the horizontal dashed lines meant to signify? Could the authors provide more examples of "alleviation" and add examples of "aggravation" as well (pertaining to major concerns# 3)?
4. How Bayesian analysis is used in ERG analysis is unclear and should be justified better with some intermediate (between raw ERG and final process) results. Why is variance in ERG of particular concern while all other data in this study also come with large variability? Couldn't single ERG recordings not be used, like single fly behavioral timeseries?
5. Suppl Fig 2. The authors report seizure -like activity. The methodology used to define a seizure was unclear. Is it based on previously published methods? Are the seizures only occurring after a mechanical trigger (vortex)? Do the flies exhibit any involuntary movements? What about after L-Dopa administration? Supplementary video recordings and an expanded description of the assay would be helpful.
6. Suppl Fig. 6. By looking at gene expression profiles in the brain, the authors conclude that clustering of gene expression cannot explain clustering of mutants according to behavior. Since PD-important genes are likely expressed widely in the fly, can the authors comment on the possibility of linking genes to behavior by examining expressions outside the brain as well? For instance, the gut (and gut-brain connection) has received particular attention in several Parkinson's studies.

Reviewer #5

(Remarks to the Author)

Version 1:

Reviewer comments:

Reviewer #1

(Remarks to the Author)

The revisions, including new experimental data, have resulted in a much improved manuscript. Collectively, a remarkable amount of work is presented, making for an important and valuable contribution. It appears that a technical glitch remains with accessing Supplemental Table 3.

Reviewer #2

(Remarks to the Author)

This manuscript provides functional analyses of mutations in Drosophila homologs of 24 genes that have been linked to Parkinson and other motor dysfunctions in humans. They use multiple assays to cluster the phenotypes of these mutants and show that the genes fall into two broad groups (or 3 groups depending on how one sets the cutoffs). Remarkably, orthogonal assays lead to the same basic groupings; even more remarkably, response to pharmacological treatment largely obeys the same groupings. Perhaps most striking, this is true not only of genes that are known to have related functions, but even of genes that seem superficially to have little in common but that co-cluster in the unbiased assays and then are rescued by the same drugs.

This paper brings an unexpected and long-sought order to what has been a bewildering multiplicity of functions and properties of PD-associated genes. It also offers a whole new way to think about the basic pathogenesis of PD, a meaningful way to segregate pathological genes into functional pathways, and a model for how we can dissect other diseases with complex genetic etiology, including, AD, FTD-ALS, and others. It is of huge significance to the specific field of Parkinson disease, to the larger field of human neurodegenerative disease, and in the broadest sense as a model for dissection of complex genetic processes.

I found the original submission of this manuscript to be quite impressive; the revised version is even better. In my opinion the authors are to be applauded for the clarity and thoroughness with which they addressed the comments of all the reviewers, myself included. I am entirely satisfied by their responses. There were a number of referee requests to expand on details of the clustering methods, which clearly were central to the authors' conclusions; I found the responses to be sensible and convincing.

One thing that struck me in reading the comments is that a number of the referee concerns boil down to the question of why assays like ERG and sleep, that seem to be only peripherally related to diagnostic phenotypes of "parkinsonism", should be appropriate tools for clustering molecular lesions associated with motor dysfunction. However, it seems to me that the success of these measures (as demonstrated by the concordance between clustering by multiple assays and by response to treatment) is one of the most profound results of the entire project. It provides an astonishing demonstration that the disease pathogenesis is rooted, not in the details of dopamine biology per se, but in basic mechanisms of neuronal cell biology. These data provide powerful support for the idea that it is the selective sensitivity of particular neuronal cell types to specific aspects of that basic cell biology that distinguish among the various neurodegenerative diseases (an idea that is also underscored by results in a previous study from this group, Prasher, 2023). The work by these authors now gives us a way to approach the basis of that selective sensitivity that has been so elusive.

Reviewer #3

(Remarks to the Author)

I am fully satisfied with the authors' responses and additional experiments, and I have no further questions. The authors have appropriately addressed the reviewers' questions and critiques, and this revision is sufficient for publication.

Reviewer #4

(Remarks to the Author)

The investigators have provided a substantially improved revised manuscript. The changes to Figure 3 and its related discussion offer greater clarity regarding both the methodological approach and the conclusions drawn regarding the genetic interaction screen using ERGs. Inclusion of the Calinski-Harabasz index bolsters their earlier claims about data clustering. Revised Supplementary Figures 6 and 8 are helpful. The revised Supplementary Figure 3 along with newly included videos capturing seizure-like behavior is a valuable addition, that enables a more comprehensive assessment of the described phenotype. Moreover, the incorporation of small molecule treatment for all mutants, as detailed in revised Figure 4, provided compelling evidence to support the assigned classifications. I still find description of fly speed calculations unnecessarily complicated but that may be more due to the work of Geissmann et al rather than the current manuscript.

The revised manuscript is recommended for publication.

Reviewer #5

(Remarks to the Author)

We thank the reviewers for their thoughtful questions and remarks. We have addressed the comments and below we summarize the most important additions:

- 1) Our work revealed different groups of familial Parkinson's disease and related genetic causes of parkinsonism based on behavioral screening. We conducted several additional analyses to verify the robustness of our clustering. *First*, we verified that the parameters we included for clustering are independent from one another by conducting a correlation analysis. This shows that the (independent) parameters we used for clustering do not significantly correlate; *Second*, we removed single parameters from our clustering analysis and find this does not majorly affect the final clustering result; *third*, we included the *pdf* mutant. We previously found that *parkin*, *pink1* and *pdf* mutants show similar sleep phenotypes¹ and now find in our unbiased assessment that *pdf* clusters with *pink1*, *parkin* and *chchd2* in "group B". Taken together, these new experiments further support the robustness of our clusters.
- 2) We tested if all mutants in each of the groups with a dopaminergic synaptic innervation defect are rescued by Q10 (that boosts mitochondrial function) or R55 (that is a retromer chaperone). We show that the synaptic innervation of dopaminergic neurons of mutants of the group that contains *pink1* and *parkin* (known to affect mitochondria) (group B) are rescued by Q10 but not by R55, and vice versa, that the synaptic innervation of dopaminergic neurons of mutants of the group with *iPLA2VIA* and *vsp35* (known to affect the retromer) (group A1-3) are rescued by R55 and not by Q10. These results are consistent with our conclusion that the genetic space of Parkinson's disease and related genetic causes can be 'stratified' in specific groups each with specific molecular dysfunction.
- 3) Some of our mutants showed activity-induced seizure-like behavior. We show this is not rescued by L-DOPA (while the dopaminergic neuron-related phenotypes are rescued).
- 4) We included RT-PCR, validating our mutants. We also noted problems and molecular genetic rearrangements in our *park*^{KO-WS} mutant that we still do not fully understand. We therefore replaced *park*^{KO-WS} throughout the paper by *park*^{1/Δ21} (a null mutant; also confirmed by RT-PCR). This includes the initial phenotyping: SING, TH-labeling and L-DOPA responsiveness, as well as the ethoscope recordings, ERG-genetic interactions against the rest of the mutants, clustering and the Q10 and R55 rescue experiments. We find the new *parkin* mutant -as expected- clusters with *pink1* and *chchd2* (also thought to affect mitochondria) and it is rescued by Q10 (and L-DOPA), but not by R55.
- 5) We expanded the expression analysis of mutants to include single cell sequencing datasets of the entire fly body. Like our analysis of gene expression across brain cell types, we also do not find a correlation between expression pattern and the assignment of mutants to groups in our clusters. In other words: gene expression patterns are likely not sufficient to define the 'Parkinson's-disease-and-related-genetic-causes-of-Parkinsonism' groups.

Below we address each reviewer point one-by-one.

REVIEWER COMMENTS

Reviewer #1 (Remarks to the Author):

Kaempfer et al. have performed a targeted screen using *Drosophila* to systematically study several dozen genes implicated in PD and other genetic/neurodegenerative disorders that cause parkinsonism. They characterize locomotor impairment, recovery from vortexing, dopaminergic neuron loss, sleep, and

electroretinograms. They classify genes into several groups based on the pattern of phenotype observed. They also interrogate pairwise genetic interactions to define potential pathways, and test drugs predicted to modulate mitochondrial energetics and vesicular trafficking. Overall, this work provides a valuable compendium of reagents and data for further cross-species genetic dissection of familial forms of parkinsonism, and begins to define genetic subgroups sharing common molecular mechanisms.

Please consider the following suggestions for improving the manuscript:

-I believe that the term “parkinsonism” needs additional qualification for use throughout this manuscript. “Parkinsonism” refers to a rather non-specific human motor syndrome—a pattern that usually localizes to basal ganglia dysfunction. This is similar to other human neurologic motor syndromes, such as “spastic paraparesis” or “ataxia”, which usually result from upper motor neuron or cerebellar dysfunction, respectively. Thus, parkinsonism has anatomic localizing value but does not imply a specific molecular etiology. Drugs / toxins, vascular lesions (stroke, bleeds), and even brain tumors can also cause parkinsonism. This manuscript is focused on genetic/neurodegenerative causes of parkinsonism rather than “parkinsonism” more generally. This is where the *Drosophila* model really is most helpful due to genomic conservation to examine consequences of manipulation of gene homologs for dopaminergic neuronal health. By contrast, there are no basal ganglia in the invertebrate nervous system—making it a poor model to examine higher-level circuit level contributions to parkinsonism. I would therefore recommend that this be better explained in the introduction. Perhaps clarify that the most immediate interest is in “Parkinson’s disease and related genetic/familial causes of parkinsonism”. The discussion of non-motor manifestations (Lines 56-57) is also confusing, and perhaps a distraction—consider deleting. These features are not part of “parkinsonism”, nor do they localize to dopaminergic neuronal / basal ganglia dysfunction, but may be seen accompanying parkinsonism in PD. The term is less relevant in thinking about other non-PD forms of parkinsonism. In the discussion, “the specific causes of most idiopathic parkinsonism cases remain unknown” (line 397)—Are the authors intending here to mean idiopathic PD? This would make the most sense. Similarly, I recommend revising the text later in this paragraph (lines 405-409): “...capture most of the molecular complexity underlying all parkinsonism cases”. For clarity, it would best to specify “PD and related genetic causes of parkinsonism”, rather than other causes of parkinsonism, such as cerebrovascular pathology.

We thank the reviewer for the comments and have now clarified the terminology of parkinsonism in the manuscript. We indeed address familial Parkinson’s disease and genetic forms of parkinsonism in the manuscript and, therefore, defined the term “parkinsonism” in the introduction as “Parkinson’s disease and related genetic causes of parkinsonism (for simplicity referred to as “parkinsonism” hereafter)”. We have shortened the description of non-motor manifestations in the introduction as well. We have further addressed the phrases in the discussion and rephrased idiopathic parkinsonism to “idiopathic PD” and “familial PD and genetic causes of parkinsonism”.

-Very limited information is provided on how genes were selected? Were there specific search criteria or was this based on certain review articles? The diseases under consideration are rather heterogeneous, including both pediatric and adult conditions. Some are considered primary dystonias, in which parkinsonism is a secondary or a more variable feature. Given the importance of gene selection to the design of the study, I would consider including a descriptive table in the main text, perhaps a condensed version of Supp Table 1. I think it is also important to mention what (if any) human genes were excluded due to poor conservation in *Drosophila*. It is useful to classify the different diseases being considered under the umbrella of genetic causes of parkinsonism, such as Parkinson’s

disease (GBA, LRRK2), autosomal recessive juvenile parkinsonism, dystonia-parkinsonism, NBIA, etc. I'm not certain that the distinction of typical vs. atypical in Supp. Table 1 is particularly useful—I would consider most of the genes/disorders being included here as “atypical” causes of parkinsonism, with PD being the most common cause of neurodegenerative parkinsonism, and therefore “typical” (e.g., GBA, LRRK2). Some of the genes included, such as COQ2, NUS1, eIF4G1 and HTRA2/Omi are controversial with limited replication or non-replication. It may be reasonable to include them as part of the screen, but it would be best to note the caveats. If these are indeed not true genetic causes of parkinsonism, it could skew / bias the clustering.

We curated a comprehensive list of PD and related genetic causes of parkinsonism based on (1) clinical insights obtained through collaboration with neurologist Prof. Dr. Wim Vandenberghe (UZ Leuven) and (2) an extensive review of the literature. We have added this information in the results section. We also indicated in Supplementary Table 1 where further replication in patients is awaiting. Furthermore, we added in Supplementary Table 1, that *SNCA* was excluded from the list, because it is not conserved in the fly genome, and included descriptions of the types of disease caused by mutations in the listed genes as requested by the reviewer. Finally, we also included aspects of this information in condensed form in Reviewer Fig.1a and Fig.1a.

Figure 1

Reviewer Fig.1 (Fig.1): PD and related genetic causes of Parkinsonism mutants show age-dependent and dopamine-dependent phenotypes. Related to Supplementary Fig.1-3 and Supplementary Table 1,2.

(a,b) Quantification of startle-induced-negative geotaxis (SING) at 6±1 d after eclosion (young, (a)) and 42±2 d after eclosion (aged, (b)) of parkinsonism mutants normalized (norm) to the control (CS^{w1118}^{w+},

dark grey). Due to their decreased survival *nutcracker*^{KO-WS} were tested at 15 d, *Pink1*^{KO-WS} at 22±2 d, *iPLA2-VIA*^{KO-WS} and *gba*^{KO} mutants at 25±1 d and included in the “aged” analysis. The “/+” refers to heterozygosity and “/y” to hemizyosity. Turquoise colored bars represent p<0.05, grey colored bars represent p>0.05, ANOVA Kruskal-Wallis with Benjamini-Hochberg. Bars: mean ± SD; points represent groups of animals and (a) N≥4 and (b) N≥7. (c) Quantification of dopaminergic synaptic area within mushroom body (MB) area in aged mutants (as in (b)) relative to the control. Colored bars represent p<0.05, ANOVA Kruskal-Wallis with Benjamini-Hochberg. Bars: mean ± SD; points are individual animals N≥9 per genotype. (d) Maximum projection confocal image of *control* and *Pink1*^{KO-WS} fly brains (young: 5±1 d and aged: 22±2 d) stained with anti-TH (cyan) and anti-DLG (magenta) antibodies, where DLG marks the post-synaptic site of MB. The black and white images represent the thresholded TH area (in white) of middle z-plane within the region of interest (ROI, yellow, outline of MB). Scale bar: 20 μm. (e) Linear regression plot with 95% confidence intervals (grey area) of the mean normalized SING score and mean normalized TH positive area within MB area of aged parkinsonism mutants show positive correlation with R²=0.53, p=0.0001. (f) SING quantification of aged flies (as in b) treated with L-Dopa or D-Dopa (control, grey bars) 10 d prior to the assay relative to D-Dopa treated *control*, Two-way ANOVA with Tukey’s multiple comparison, significant p values are indicated in pink. Bars: mean ± SD; points represent groups of animals and N≥9. ADP, adult-onset dystonia-parkinsonism; AD, autosomal dominant; AD PD, autosomal dominant Parkinson’s disease; APD, adult-onset Parkinson’s disease; AR, autosomal recessive; DLG, Discs-large; DP, dystonia-parkinsonism; DRD, DOPA-responsive dystonia; EPD, early-onset Parkinson’s disease; GD, Gaucher disease; h, human; JPD, juvenile-onset Parkinson’s disease; MB, mushroom body; mod PD, modifier for Parkinson’s disease; MSA, Multiple system atrophy; PPS, parkinsonian-pyramidal syndrome; SING, startle-induced negative geotaxis; SNDC, childhood-onset striatonigral degeneration; TH, tyrosine hydroxylase; WS, white-STAR; XLR, X-linked recessive; XPD, X-linked parkinsonism.

-Related to the above, I believe it is incorrect to talk about a fly recapitulating “hallmarks of parkinsonism”, since flies do not have basal ganglia, and except for movement speed, other essential features that define parkinsonism (rigidity, tremor, postural reflexes) are of course not assessed. Perhaps note instead that flies recapitulate “dopamine responsive, progressive motor impairment”. We thank the reviewer for the comment and have specified the headline as “Parkinsonism fly collection recapitulates dopamine responsive, progressive motor impairments”.

-Based on Fig1a, it appears that different genes are being tested in heterozygosity versus homozygosity (or hemizyosity), with most being tested as heterozygotes? What is the rationale / justification? Was this based on inheritance model? This does not seem to be the case. For example, ATP13A2 causes recessive parkinsonism, but the fly mutant, anne, is tested in heterozygosity? GBA on the hand usually is associated with PD in heterozygosity, but it appears that homozygous KO is tested. What about embryonic or adult lethality?—this information does not appear to be provided anywhere. We used for our analysis the strongest loss of function situation that was still viable, enabling us to compare the reduced function phenotypes of the relevant genes. The majority of mutants is embryonic lethal when homozygous necessitating the use of heterozygous flies. We have added a Supplementary Table 2 (Reviewer table 1) describing the embryonic lethality, fertility and oldest age at which we tested the mutants.

Reviewer Table 1 (Supplementary Table 2): Overview of parkinsonism fly line characteristics, related to Fig. 1.

Fly lines	Viability		Fertility		Max age tested (days after eclosion)
	het	hom	het	hom	
anne ^{KO-WS}	A	<L3	+	n/a	45
VhaM8.9 ^{KO-WS}	A	<L3	+	n/a	45
Chchd2 ^{KO-WS}	A	A	+	+	45
Coq2 ^{KO-WS}	A	<L3	+	n/a	45
DJ-1a/b ^{KO-WS}	A	A	+	+	45
Rme-8 ^{KO-WS}	A	<L3	+	n/a	45
auxilin ^{KO-WS}	A	<L3	+	n/a	45
eIF4G ^{KO-WS}	A	<L3	+	n/a	45
omi ^{KO-WS}	A	A	+	-	45
Lrrk ^{KO-WS}	A	A	+	+	45
park ^{1/Δ21}	A	A (transhet)	+	- (Pesah et al. 2004)	45
Pink1 ^{KO-WS}	A	A (hemi)	+	- (hemi)	25
iPLA2-VIA ^{KO-WS}	A	A	+	+	27
Rab39 ^{KO-WS}	A	A	+	+	45
Synj ^{KO-WS}	A	<L3	+	n/a	45
Vps35 ^{KO-WS}	A	<L3	+	n/a	45
loqs ^{KO-WS}	A	<L3	+	n/a	45
nutcracker ^{KO-WS}	A	A	+	-	15
VAC14 ^{KO-WS}	A	<L3	+	n/a	45
Vps13 ^{KO-WS}	A	<L3	+	n/a	45
Punch ^{KO-WS}	A	<L3	+	n/a	45
Gdh ^{KO-WS}	A	<L3	+	n/a	45
gba ^{KO}	A	A	+	+	27
Tango14 ^{KO-WS}	A	<L3	+	n/a	45

A, adulthood; hemi, hemizygous; het, heterozygous; hom, homozygous; L3, 3. larval stage; n/a, not applicable; transhet, transheterozygous; +, fertile; -, sterile; bold, genotype used for analysis.

-lines 160-171: seizure is not a common accompaniment of PD and neurodegenerative parkinsonism. Moreover, “seizure-like behavior” following vortexing of flies is not necessarily representative of the pathophysiology underlying an epileptic seizure—caveats should be noted. Why is this considered a “seizure” assay, rather than recovery from a kind of trauma—head trauma is also a cause of parkinsonism...

Based on our literature research epileptic seizures are a comorbidity in patients with PD and parkinsonism². However, it is indeed not a common symptom, and described in (rare) patients with mutations in *Synj*³, *DNAJC6*⁴, *DJ1*⁵, *ATP6AP2*⁶, *Rab39B*⁷, *PLA2G6*⁸ and *gba*⁹ (references at the end of this response letter).

We acknowledge differences between the pathophysiology of epileptic seizures in humans and seizure-like behavior in *Drosophila melanogaster* and made this clear in the text. We also refer to the observed phenotype in flies strictly as “seizure-like behavior”, terminology commonly used in this field to describe this phenotype, which is likely the consequence of underlying defects in synaptic transmission.¹⁰⁻¹²

During our work we noted unintentionally that mechanical stimuli induced episodes in some of our mutant fly lines and this prompted us to systematically investigate this “seizure-like behavior” using this well-established and validated assay. Previous work had indeed already shown that flies with mutations in different ion channels, the homologues of which are associated with epilepsy in humans,

and flies with (subtle) mutations in synaptic transmission genes, the homologues of which are also associated with epileptic problems in humans, show defects in this assay. Furthermore, similar seizure-like responses were also seen in flies following electroconvulsive stimulation and several compounds are known to induce seizures (epileptiform activity) in humans and seizure-like activity in *Drosophila*, often via conserved mechanisms affecting ion channels, neurotransmitter systems, or metabolism, indicating that multiple modalities of neuronal overstimulation can produce comparable phenotypes in sensitized fly strains (e.g. Picrotoxin, Veratridine, 4-Aminopyridine, etc.)^{10–19}

While not central to our work, we wanted to note in the manuscript the correlation between observations of seizures in patients and the response of specific fly mutants with homologous mutations in this assay.

-lines 272-283: It is possible that genes participate in common pathways, yet may not be expressed in the same cells but rather in the same multicellular functional circuits comprised of distinct cell types?? Perhaps consider this as a discussion point.

We agree with the possibility that genes contributing to shared pathways may not necessarily be co-expressed within the same cell type but could instead act within multicellular functional circuits comprising distinct cellular populations. We are discussing this in the new manuscript.

-Fig 3b: why was ERG rather than locomotor testing used for the gene-gene interaction analysis? Some additional justification/rationale is needed here? A preponderance of genes have ERG deficits in heterozygosity—is this result expected? It would be interesting to test whether any of the interactions are consistent using the SING assay.

1) ERGs are a very sensitive and quantitative readout and ideal for semi-high-throughput screening of the 276 gene pair combinations. While the sensitivity of the ERG assay enabled the detection of phenotypes in young and heterozygous flies, SING assays do not have the same throughput nor the needed sensitivity.

2) To address the second part of this question, we selected 7 genetic interaction-pairs from the ERG screen (Reviewer Fig.2a) and tested the decline of TH+ synaptic area in the Mushroom Bodies of aged flies. We chose the TH+ area assay instead of the SING assay, because TH+ area assay is experimentally more feasible and the outcome of both TH+ synaptic area and SING assays is highly correlated (Fig 1). Our results show that the gene pairs of *Synj*^{KO-WS/+} and *iPLA2-VIA*^{KO-WS/+} as well as *Synj*^{KO-WS/+} and *VhaM8.9*^{KO-WS/+} (Reviewer Fig.2g,h) show a positive genetic interaction strength of 0.21 and 0.19, respectively, similar to the interactions we measured with ERGs. *Vps35*^{KO-WS/+} and *omi*^{KO-WS/+} also showed genetic interaction, the direction of which was opposite to the genetic interaction we measured with ERGs. Finally, *Pink1*^{KO-WS/y} and *nutcracker*^{KO-WS/+}, *VAC14*^{KO-WS/+} and *auxilin*^{KO-WS/+}, *Rme-8*^{KO-WS/+} and *VAC14*^{KO-WS/+} as well as *gba*^{KO-w/+} and *VhaM8.9*^{KO-WS/+} showed a genetic interaction in the ERG assay but not in the TH+ labelling assay. Hence, we are able to recapitulate some of the genetic interactions across both assays but not others (Reviewer Fig.2b-h). The neuronal substrate underlying each assay, the sensitivity of each assay, and the different experimental time points used for measurements (young vs aged flies) could be reasons for the specific genetic interaction effects of our mutants across the different assays. This is not per se unexpected: one example from *Drosophila* is the Notch-EGFR relationship that is explicitly context-dependent; cooperative in some tissues and antagonistic in others (e.g., *argos* regulation differs between adult muscle progenitors and the wing)²⁰. We show these results as a reviewer figure below but did not include them in the main manuscript (we can do so upon the reviewer's request).

Reviewer Fig 2: The genetic interaction of the ERG screen cannot consistently be reproduced using the TH+ area assay in aged flies. (a) Overview of tested interactions marked in the heatmap of Fig3b. (b-c) Gene pairs that showed aggravated GI strength in the ERG screen do not display aggravated TH+ area reduction in aged flies of *Vps35^{KO-WS/+}* and *omi^{KO-WS/+}* as well as *Pink1^{KO-WS/y}* and *nutcracker^{KO-WS/+}* flies. (d-h) Only 2 out of 5 tested gene pairs show a positive GI strength in the TH+ area of aged flies: *Synj^{KO-WS/+}* and *iPLA2-VIA^{KO-WS/+}* as well as *Synj^{KO-WS/+}* and *VhaM8.9^{KO-WS/+}*. Aged flies of *VAC14^{KO-WS/+}* and *auxilin^{KO-WS/+}*, *Rme-8^{KO-WS/+}* and *VAC14^{KO-WS/+}* as well as *gba^{KO-w/+}* and *VhaM8.9^{KO-WS/+}* show no genetic interaction in the TH+ area assay.

-lines 317-319: I would recommend softening the claim that behavior and gene interaction analysis using ERG “define the same 3 parkinsonism subgroups”. While modest overlap was found, clearly the classifications also diverge in important ways.

We have rephrased it in the text: “Thus, both the unbiased behavioral analyses and the GI profiles define similar parkinsonism subgroups, which ultimately define shared molecular handles into different disease subtypes.”

-340-358: Were additional genes tested with drug feedings for rescue within each group (A, C) besides the examples shown?

We have extended the treatment with the small molecules Q10 and R55 for all the mutants with a significant dopaminergic innervation defect in aged flies (Fig.1c). We tested whether the pharmacological feeding affects the area of TH+ labelling and included the results in Reviewer Fig.3, the new Fig.4 and Supplementary Fig.12. Interestingly, Q10 rescues the TH+ innervation defect of mutants in group B (both B1 and B2) (*Pink1*^{KO-WS}, *park*^{1/Δ21}, *anne*^{KO-WS/+}, *auxilin*^{KO-WS/+} and *nutcracker*^{KO-WS}) and some mutants of the “light blue” A3 subgroup (*VAC14*^{KO-WS/+}, *VhaM8.9*^{KO-WS/+}, *Gdh*^{KO-WS/+} and *eIF4G*^{KO-WS/+}). In contrast, Q10 does not rescue the TH+ innervation defect of groups A1 and A2 (dark and medium blue labels). *Tango14*^{KO-WS/+} in group A2 shows even a worsening of the phenotype.

R55 treatment significantly rescues the dopaminergic synaptic innervation defect in aged mutants of groups A1 and A2 and one mutant of A3 (*iPLA2-VIA*^{KO-WS}, *DJ-1a/b*^{KO-WS}, *gba*^{KO}, *Rab39*^{KO-WS}, *omi*^{KO-WS}, *Vps13*^{KO-WS/+}, *Synj*^{KO-WS/+}, *Tango14*^{KO-WS/+} and *VhaM8.9*^{KO-WS/+}), while members of group B are not rescued by R55. A Fisher’s exact test revealed a significant association between “group” and “small-molecule treatment” ($p < 0.001$), further supporting our conclusion that dopaminergic defects originate from cluster-specific pathomechanisms. These results were included in the reworked manuscript.

Figure 4

Reviewer Fig.3 (Fig.4): Dopaminergic neuron defects are rescued by subgroup specific treatments. Related to Supplementary Fig.10. (a) Maximum projection confocal image of aged control (*CS^{w1118}w+*), *Pink1*^{KO-WS}, *iPLA2-VIA*^{KO-WS} fly brains treated with 1 mM Q10 or solvent-control (ethanol) stained with anti-TH (cyan) and anti-DLG (magenta) antibodies, where DLG marks the post-synaptic site of MB. The

black and white images represent the thresholded TH area (in white) of “middle z-plane” within the ROI (yellow, outline of MB). Scale bar: 20 μ m. (b) Quantification of dopaminergic synaptic area within MB area in aged mutants treated with Q10 (white bars) or control treatment (filled bars) relative to control with solvent treatment. Points are individual animals $N \geq 9$ per genotype. Bars: mean \pm SD; Two-way ANOVA with Tukey’s multiple comparison, significant p values are indicated with * $p < 0.05$, ** $p < 0.01$, *** $p < 0.001$, **** $p < 0.0001$. (c) Maximum projection confocal image of aged *control*, *Pink1^{KO-WS}*, *iPLA2-VIA^{KO-WS}* fly brains treated with 12 μ M R55 or solvent-control, H₂O, labeled with anti-TH (cyan) and anti-DLG (magenta) antibodies and thresholded TH+ area of “middle z-plane” within the ROI (yellow, outline of MB). Scale bar: 20 μ m. (d) Quantification of dopaminergic synaptic area within MB area in aged mutants treated with R55 (white bars) or control (filled bars) relative to controls with solvent treatment. Points are individual animals $N \geq 13$ per genotype. Bars: mean \pm SD; Two-way ANOVA with Tukey’s multiple comparison, significant p values are indicated with * $p < 0.05$, ** $p < 0.01$, *** $p < 0.001$, **** $p < 0.0001$. # represent significant reduction $p < 0.05$.

-As the authors are aware, the main pathways / gene clusters, such as mitochondrial quality control and endocytic trafficking are well established, and this provides an opportunity to show how the method applied re-demonstrates known relationships among genes. Thus, the classification of genetic causes of parkinsonism into groups would be most compelling if it immediately allowed novel predictions that could be validated from human data. While it may not be reasonable to include this in the current manuscript, maybe the authors speculate on what type of prediction might be made for consideration in future analyses? Could this guide gene discovery or therapy? One potential future approach to consider in the discussion is to apply the method to candidates from PD GWAS, where dozens of potential genes might be assigned to the clusters defined from Mendelian forms of PD / parkinsonism.

We strongly agree with the reviewer and are grateful for highlighting the potential of our work. Indeed, as a first step the subgroups can be used to identify commonalities in clinical data of parkinsonism patients with genetic causes. Those clinical characteristics could then be used to associate idiopathic cases to subgroups. These patient subgroups could help with clinical trial design, biomarker discovery and could unmask clinical benefits of subgroup-specific treatments. Moreover, candidates of GWAS studies and new risk factor genes with possibly unknown function could also be modelled in flies and assigned to one of our subgroups. We have highlighted these future directions in the discussion.

Some addition minor questions / suggestions:

-Lines 111-113: the description gives the impression that the SING phenotype may be specific for DA neuronal defects; however, many other CNS (or PNS) lesions can cause SING phenotypes.

Yes, this is correct: the PAM subcluster of DAN is necessary for SING, but not sufficient^{21,22}. Without excluding contributions of other parts of the brain, our observation of a significant correlation between TH+ area reduction in the Mushroom Bodies and a SING defect in our mutants (Fig. 1e) suggests that at least PAM DAN are involved. We have adapted the text to “SING defects in parkinsonism fly models have previously been shown to involve dopaminergic impairments²⁶⁻²⁸”.

-Fig 1: rather than “young” and “aged”, it would be best to note the exact age ranges in the fig. For the aged flies in our experiments, we pooled flies in a given age range, usually 42 \pm 2 d after eclosion, but for some genotypes we needed to divert from this: nutcracker^{KO-WS} at 15 d, Pink1^{KO-WS/y} at 22 \pm 2 d, iPLA2-VIA^{KO-WS} and gba^{KO} mutants at 25 \pm 1 d. Adding age ranges in the figure would make the x-axis

labeling rather bulky. Therefore, we summarized the age range as “aged” in the figure and describe the individual ages of “aged” flies in the figure legend. For consistency we labelled the 6+/-1 d old flies as “young” flies.

-lines 155-157: perhaps note that only “selected mutants” were tested with H2O controls (Supp Fig 1g)

Thank you for pointing this out. We added the information in the text: “We do not observe such rescue when flies are fed the stereoisomer D-Dopa (Fig.1f) or H2O (solvent control, for selected mutants, Supplementary Fig.2f)”.

-lines 184-185: it may be helpful to many readers to include additional discussion of the sleep disruptions that occur in PD/parkinsonism and what is known about anatomy / pathophysiology to put the screening based on this phenotype into a bit better context. We added examples of observed sleep defects in disease: “Sleep is an evolutionary conserved behavior^{23,24} and various aspects of the sleep cycle are affected in the prodrome of parkinsonism such as insomnia, excessive daytime sleepiness, and other alterations^{25,26}.”

Specific sleep disturbances as observed in patients, such as REM sleep behavior disorder (RBD) or restless legs syndrome (RLS), cannot be reliably assessed in *Drosophila* due to fundamental differences in neural circuitry and behavioral manifestations. Instead, we used sleep, as part of a broader set of circadian and activity-related parameters, as a quantitative behavioral readout to identify subgroups of parkinsonism-models; the genetics affected in each group then enabled us to reveal the group-specific underlying molecular mechanisms.

-line 352: “We challenged this result”--unclear what this phrase means?

We have rephrased it in the reworked manuscript.

-lines 359-362: these final sentences seem better suited to discussion. More context and citations are suggested for the claim that group B is related to autophagy/proteostasis.

We have adjusted the discussion section with an expanded explanation and more citations.

Reviewer #2 (Remarks to the Author):

The manuscript by Kaempfer, et al. is extremely impressive and exceptionally significant. The authors generate comparable knockout mutations in 24 genes in *Drosophila* that are orthologous to genes strongly associated with Parkinson Disease in humans. They then performed unbiased assays investigating a variety of behavioral phenotypes of these mutant flies and using them to cluster the mutations into groups, as well as systematically performing a genetic interaction test among the mutations based on neuronal function. Quite remarkably, the gene clusters suggested by the multidimensional behavioral analysis show very good concordance with the pattern of genetic interactions, suggesting that the two orthogonal approaches are clustering PD risk genes consistently into the same three discrete classes. Finally, a pharmacological test provides an initial suggestion that the classes defined by phenotype and genetic interaction may also serve to stratify individual mutations into classes that are differentially sensitive to particular drug treatments.

A large number and variety of genetic loci have been linked to PD, as is also true of other neurodegenerative diseases. Like those other diseases, while the genetics have highlighted a variety of molecular pathways as potentially being implicated in PD it has been exceedingly difficult to see an underlying order in that diversity of genetic causes and predisposing factors in PD. The current work potentially provides a unique insight into the genetic structure of the disease by giving unbiased,

systems-based grounds for pulling together different genetic loci into a small, defined collection of experimentally-motivated groupings. In addition to the value of the specific results, it provides a template that can and should be used with other datasets to test and expand the analysis here.

I have one more significant comment and one or two minor comments

1. I was disappointed that results of the pharmacological test (Fig 4) were reported for only 4 of the 24 mutations. The concordance of treatment response with gene groupings is certainly suggestive, but the number of genes tested is sufficiently limited to make one worry that this is just happenstance. If possible, it would certainly be preferable to see the results of these pharmacological tests across the whole collection of mutants for at least one of the two assays of Fig 4, if not for both. One doesn't expect to see perfect agreement, just as the concordance between the behavioral and ERG assays is not perfect and is not expected to be, but as it stands it is not clear how strongly to interpret this result. This is an excellent suggestion and we have now extended the treatment with Q10 and R55 across the entire mutant collection, scanning for rescue of the dopaminergic innervation defect in aged flies (Reviewer Fig.3 above in the response to reviewer 1 and new Fig.4 and Supplementary Fig.12).

We now show that Q10 rescues the TH+ innervation defect of all mutants in group B (both B1 and B2) (*Pink1^{KO-WS}*, *park^{1/Δ21}*, *anne^{KO-WS/+}*, *auxilin^{KO-WS/+}* and *nutcracker^{KO-WS}*) and some mutants of the "light blue" A3 subgroup (*VAC14^{KO-WS/+}*, *VhaM8.9^{KO-WS/+}*, *Gdh^{KO-WS/+}* and *eIF4G^{KO-WS/+}*). In contrast, Q10 does not rescue the TH+ innervation defect of groups A1 and A2 (dark and medium blue labels). *Tango14^{KO-WS/+}* in group A2 shows even a worsening of the phenotype.

We also now show that R55 treatment significantly rescues the dopaminergic synaptic innervation defect in aged mutants of groups A1 and A2 and one mutant of A3 (*iPLA2-VIA^{KO-WS}*, *DJ-1a/b^{KO-WS}*, *gba^{KO}*, *Rab39^{KO-WS}*, *omi^{KO-WS}*, *Vps13^{KO-WS/+}*, *Synj^{KO-WS/+}*, *Tango14^{KO-WS/+}* and *VhaM8.9^{KO-WS/+}*), while members of group B are not rescued by R55. A Fisher's exact test revealed a significant association between "group" and "small-molecule treatment" ($p < 0.001$), further supporting our conclusion that dopaminergic defects originate from cluster-specific pathomechanisms. These results were included in the reworked manuscript.

2. Regarding the ERG assay, it sounds like the flies were w- at the genomic locus, with w+ introduced in the knock-in procedure (and therefore with different amounts of w+ activity in pigment cells in different lines, depending on the locus). Is that correct? At the age the assay is done, is the w+/- status of the fly expected to affect the ERG results? It would be worthwhile to address this explicitly in the text for the sake of readers who are not familiar with the assay.

This is indeed an important point that we had taken into account when designing this experiment and we have now clarified this better in the material & methods section: We crossed all heterozygous flies with *w^{118w+}* to achieve w+/w+ flies (ie wild type red pigment). The flies have additional copies of w+ as a result of our genetic manipulation, but this did not affect the appearance of the red eyes of our flies: we visually inspected the flies and our strategy indeed minimizes pigmentation variability (when inspected under the stereomicroscope). Furthermore, we were consistent in selecting age matched flies and thus limiting pigmentation differences due to age.

3. This is a very minor point, but my download of the supplements had a blank page for Suppl Table 2, the values of the genetic interaction screen.

Thank you for pointing this out. We made sure the Supplementary Table 2 (now Supplementary Table 3) is readable.

Reviewer #3 (Remarks to the Author):

The authors generated a new collection of fruit fly mutants for 24 PD-related genes and used it to group these genes into three subgroups with similar functions and behavioral activities. Since *Drosophila* has been used very efficiently in previous studies to understand the pathological functions of several PD-related genes, this study can be a good starting point for understanding the molecular mechanisms of all currently known PD-related genes whose molecular mechanisms have not yet been clearly elucidated.

Major Points:

1. It is commonly accepted that disrupting the first exon of a gene typically leads to the downregulation of its expression. However, as the techniques the authors employed were novel and these flies were firstly reported in this study, the authors should provide data on the target gene expression in homozygous and heterozygous mutants to confirm that they are indeed true genetic knockouts.

We thank the reviewer for this comment and performed quantitative RT-PCR for all the mutants. The results are shown in Supplementary Fig.1 (Reviewer Fig.4). We observe in heterozygous animals marked in blue and knockout/hemizygous mutants a significant reduction in gene expression relative to the endogenous gene expression in controls.

In conducting these experiments we noticed problems with the molecular genetics of our *park*^{KO-WS} mutant, including genomic rearrangements that we still do not entirely understand. We nonetheless decided to move forward and replaced the *park*^{KO-WS} with *park*^{1/Δ21} ^{27,28} (a known null mutant that we confirm does not express Parkin by RT-PCR). We then repeated all the experiments with this new mutant, including the initial phenotyping (SING, TH-labeling and L-DOPA responsiveness, the ethoscope recordings, ERG-genetic interaction, clusterings and the Q10 and R55 rescue experiments). The results are consistent with our previous findings and show that *pink1* and *parkin* cluster in the same group that is also rescued by Q10 and not by R55. We thank the reviewer for making us go back and look at our mutants which provided the opportunity to include this new *park* mutant.

Reviewer Fig.4 (Supplementary Fig.1): Genetic strategy of Parkinsonism fly collection reduces gene expression levels, related to Fig.1. (a-a') Scheme of the genetic approach for the generation of the

parkinsonism mutant fly collection. The first common exon of all possible *Drosophila* transcripts of the targeted parkinsonism gene was replaced by attP-flanked mini-*white* gene, by homology directed repair with the pWhite-STAR (a) or pWhite-STAR2 (a') using CRISPR/Cas9, creating a null mutant. *I-SceI* sites were utilized in the rare event of full donor plasmid integration (methods). For *park* we used *park*^{1/Δ21}. (b) Relative gene expression levels, measured by quantitative RT-PCR in young flies with primers to the indicated gene (left) in controls and parkinsonism mutants. Data are expressed as a relative value compared to the endogenous *Drosophila* gene expression. The “/+” refers to heterozygosity (genotypes indicated in blue) and “/y” to hemizyosity. Unpaired t-test; Bars: mean ± SD; points are samples consisting of groups of animals N≥1.

2. Measuring TH+ area relative to the mushroom body area is a very interesting and effective approach. Additionally, the observation that this index is primarily age-dependent and correlates with motor activity is intriguing. However, it is important to note that the main function of mushroom body in fruit fly is not closely related to the main symptoms of Parkinson's disease (parkinsonism). Clarifying this point would help readers better understand the relevance of this new readout.

We agree with the reviewer and clarified this in the text. We used the area of the mushroom body only as a measure of the individual mutant brain size to normalize the TH+ synaptic area of PAM dopaminergic neurons projecting onto the MB. Thus, this is a cell autonomous read-out of dopaminergic synapse integrity, which correlates with the behavioral (i.e. functional) readout: the SING assay and the underlying circuitry with dopaminergic neurons projecting onto the MB have been characterized in the past^{21,22}.

3. The authors compared parkinsonism phenotypes in Figure 1 using only genetic knockout flies, which is a logical approach. However, in subsequent figures, such as Supplementary Figure 2, where seizure phenotypes are addressed, the authors clustered PD genes using data from both humans and flies. Therefore, it is crucial to consider whether the mutations in PD genes result in loss-of-function or gain-of-function effects. For example, since pathogenic mutations in LRRK2 are known to be gain-of-function, it would be more appropriate to use LRRK2 gain-of-function mutants, such as G2019S mutant, for a more accurate analysis.

In this large study we decided to use a consistent genetic approach, but it will be interesting in future work to use actual pathogenic mutants. Here our goal was to use the strongest loss-of-function mutants that we could generate and then compare the genetic factors to one another without confounding our analysis at this stage with the effects of specific pathogenic mutations/mechanisms.

4. The connection between the ERG/sleep phenotypes and MB TH+ area is ambiguous. It would be helpful to include correlation graphs, similar to the correlation graph between SING and TH+ area in Figure 1e, showing the relationship between ERG/sleep and MB TH+ area.

To address this question we analysed the correlation of decreased TH+ area in aged flies with the ERG and different behavioral phenotypes (Reviewer Fig.5a-e). We cannot detect a general correlation in these cases. However, this analysis has limitations as it compares data of young flies (ERG and behavior phenotypes) with aged flies (TH+ area). Moreover, the ERG recordings were performed in heterozygous mutants (Fig. 3b), which are correlated in this analysis to the TH+ area of homozygous or heterozygous mutants. We have not included the figure in main manuscript but can do so upon the reviewer's request.

Reviewer Fig.5: TH+ area of aged flies does not generally correlate with ERG or behavioral parameters. Linear regression plots with 95% confidence intervals (grey area) of normalized depolarisation or behavioral parameters with normalized TH+ area/MB area (a-e). Dots represent the mean of parkinsonism mutants. R^2 and p-values are indicated.

5. This study further elucidates the genetic interactions among genes within the same subgroup. However, while the authors used chemicals such as Q10 and R55, it is important to examine whether other known genes within the three clusters have functional interactions with the PD-related genes discussed in this manuscript. For example, it would be crucial to demonstrate whether vesicle trafficking-related PD genes, such as *Synj1*, *Rab39*, *Vps13*, or *Vps35*, functionally interact with other known vesicle trafficking genes. To address this, the authors should consider selecting genes known to be involved in vesicle trafficking, such as *sec22*, and performing overexpression or knockdown experiments in a *Synj1* (or other vesicle trafficking-related PD genes) knockout background.

We thank the reviewer for the comment to further confirm the suggested molecular pathways. Rather than including an additional set of genetic interactions we extended the treatment with the small molecules Q10 and R55 for all the mutants with a significant dopaminergic innervation defect in aged flies (Fig.1c). We tested whether the pharmacological feeding affects the area of TH+ labelling and included the results in Reviewer Fig.3 (see above), the new Fig.4 and Supplementary Fig.12. Q10 rescues the TH+ innervation defect of mutants in group B (both B1 and B2) (*Pink1^{KO-WS}*, *park^{1/Δ21}*, *anne^{KO-WS/+}*, *auxilin^{KO-WS/+}* and *nutcracker^{KO-WS}*) and some mutants of the “light blue” A3 subgroup (*VAC14^{KO-WS/+}*, *VhaM8.9^{KO-WS/+}*, *Gdh^{KO-WS/+}* and *eIF4G^{KO-WS/+}*). In contrast, Q10 does not rescue the TH+ innervation defect of groups A1 and A2 (dark and medium blue labels). *Tango14^{KO-WS/+}* in group A2 shows even a worsening of the phenotype.

R55 treatment significantly rescues the dopaminergic synaptic innervation defect in aged mutants of groups A1 and A2 and one mutant of A3 (*iPLA2-VIA^{KO-WS}*, *DJ-1a/b^{KO-WS}*, *gba^{KO}*, *Rab39^{KO-WS}*, *omi^{KO-WS}*, *Vps13^{KO-WS/+}*, *Synj^{KO-WS/+}*, *Tango14^{KO-WS/+}* and *VhaM8.9^{KO-WS/+}*), while members of group B are not rescued by R55. A Fisher’s exact test revealed a significant association between “group” and “small-molecule treatment” ($p < 0.001$), further supporting our conclusion that dopaminergic defects originate from cluster-specific pathomechanisms such as “mitochondrial defects” and “endo-lysosomal trafficking-defects”. These results were included in the reworked manuscript.

Reviewer #4 (Remarks to the Author):

In this manuscript, Kaempfer and co-workers develop a new library of 24 *Drosophila* models of familial parkinsonism using CRISPR/Cas mediated knockout of homologous genes known to cause parkinsonism in humans. The authors carefully backcrossed their mutant lines for >10 generations to the same control isogenic stock. These genetically well-controlled lines are likely to be a great resource to other related studies. The authors identified three clusters of mutants that overlap in mitochondrial function, retromer/vesicle trafficking and proteostasis/autophagy. The construction of behavior-based clusters with links to common biology of a complex disease is particularly attractive, as it greatly raises

the chance that the results can potentially accelerate biomarker discovery and drug development. The results show significant reduction of climbing performance with age as well as reduction of TH positive area for most of the mutants screened. Climbing performance was ameliorated when feeding L-Dopa. Climbing and TH+ area was partially restored after feeding with subgroup specific treatments for mitochondrial dysfunction and endo-lysosomal trafficking.

The manuscript is well-written and sufficiently rigorous in most places. The studies are conducted with large numbers of animals and the methods are fairly detailed.

A few points of concern are outlined below:

Major concerns:

1. Missing alpha-synuclein staining data: The authors screened the mutants using climbing to measure motor performance and immunostaining with anti-TH (dopaminergic neurons) focusing on MB. Was there alpha-synuclein accumulation? Since these genes have been linked with other neurological disorders, including alpha-synuclein staining would be important for providing more direct conclusion as to these models recapitulating PD specific pathology.

Unfortunately, SNCA is not conserved in *D. melanogaster*. Thus, we cannot examine SNCA accumulation in the fly brain in these mutants. This does not preclude to assess the molecular genetics and the interactions between the genes homologous to those causing disease in humans. Indeed, SNCA accumulation is observed in PD and parkinsonism patients, but can also be absent in clinical PD. E.g. in familial cases with mutations in *parkin*, *Pink1*, *DJ1* and some *LRRK2* cases, post-mortem Lewy body pathology was not (always) found or only sparsely detected²⁹⁻³¹. Our goal here was to understand the very first origins of disease (that in human cells may ultimately, over the course of many years, result in alpha-synuclein pathology; also knock-in mouse models do not show such pathology). Then, based on the very first defects these animals show (including dopaminergic neuron dysfunction) build groups of common underlying dysfunction. We hope the reviewer agrees this will be valuable for the community and provide direction in (ongoing) efforts of stratification.

2. Robustness of the hierarchical classification of mutant sleep patterns: It is not clear why it was necessary to combine NMF features with user-defined behavioral features. How many clusters emerge if only NMF (more unbiased of the two) is used? In using NMF, the use of the “elbow method” is somewhat risky to this referee as this method does not have a rigorous statistical foundation and is widely considered heuristic, at best. The authors may want to consider better alternatives such as those developed by Tibshirani, Walther & Hastie (2001) and/or Calinski and Harabasz (1974). A related question: how sensitive is the clustering to the choice of the number of total features? Lastly, how do we know if the user-defined features are independent of the NMF features since they are used together?

Did the authors examine locomotor data instead of sleep? Locomotor data may contain more movement details that are “averaged out” in the process of constructing sleep timeseries.

These are all excellent questions, and we outline our response point-by-point below:

Combining NMF features with user-defined behavioral features:

To address the reviewer’s question we performed the same hierarchical clustering using only the NMF-derived features (Reviewer Fig.6b). This results in the mutants being assigned to various different groups: e.g. mutants of the original group A (Reviewer Fig.6a) end up being broadly distributed (e.g.

iPLA2-VIA, *DJ-1a/b*, *gba*, *Vps35* and *Rab39*). Similarly, mutants of group B2 end up being spread out. These differences should be interpreted in light of the complimentary nature of the NMF-derived features and user-defined features: NMF is a powerful tool for uncovering latent patterns in the temporal aspects of behavior that might be missed otherwise, but it is not sufficient to capture all relevant aspects of behavior (as we show below when we test the stability of behavior clusters). User-defined features, such as the number of sleep bouts, provide interpretable and biologically meaningful metrics that are not (fully) captured by NMF. We have not included the figure in main manuscript but can do so upon the reviewer’s request.

Reviewer Fig.6: Comparison of hierarchical clustering based on all behavior features or only NMF features. (a) Original hierarchical clustering of parkinsonism mutants based on their scaled behavior data reveal 2 behavior groups (A in blue, B in orange) with 3 and 2 subgroups each (dark and light shades, (behavior A1, A2, A3, B1, B2); $N \geq 69$. *gba*^{KO} marked in stripes as in PCA it clusters with group B (Supplementary Fig.7). (b) Hierarchical clustering of parkinsonism mutants based *only* on scaled NMF features reveals 2 groups.

Use of the elbow method:

We performed the Calinski-Harabasz Index³² and included the results in Supplementary Fig.4f (Reviewer Fig.7b). This method shows a peak at $k=2$, which would only distinguish between “sleep during typical periods” and “sleep outside of typical periods”. The next peak, $k=5$, emerges as preferable over 4 or 6 components and provides an excellent balance between more fine-grained clusters and cluster quality. Even though this method requires an assignment of the parkinsonism mutants to clusters based on their dominant NMF feature (ignoring more subtle patterns), this independent method nonetheless also identifies 5 biological meaningful patterns, similar to the elbow method. We included both the Calinski-Harabasz index result and the elbow method: Both methods agree and the elbow method is also commonly used in pattern discovery^{33,34,35}.

Reviewer Fig.7: Elbow method and Calinski-Harabasz Index support 5 behavioral patterns. (a) Elbow method supports 5 components as meaningful number of features explaining 72 % of the variance. (b) The Calinski-Harabasz Index for optimal number of clusters displays a peak at 5 components indicating that 5 clusters provide a reasonable balance between more fine-grained clusters and cluster quality.

Sensitivity of clustering to the number of features:

To check the stability of the clusters we have removed one behavioral feature or NMF component each time from the original analysis (Reviewer Fig.8a) and then compared the newly generated groups (group 1-5) to the originally determined clusters (behavior A1-B2) using the Jaccard similarity (Reviewer Fig.8c-n). By summarizing the Jaccard similarity in categories for every behavioral group (Reviewer Fig.8o), we find that most behavior groups have a Jaccard similarity higher than 0.35 (indicating substantial overlap³⁶⁻³⁸) in more than 50% of the cases when a single feature is removed. This indicates the robustness of our clustering results and the subgroups we defined. We have included this analysis in the reworked manuscript as Supplementary Fig.8.

Reviewer Fig.8 (Supplementary Fig.8): Stability of behavior-defined groups in the absence of one behavioral feature at a time, related to Fig.2. (a) Original hierarchical clustering of parkinsonism mutants based on their scaled behavior data reveal 2 behavior groups (A in blue, B in orange) with 3 and 2 subgroups each (dark and light shades, (behavior A1, A2, A3, B1, B2); $N \geq 69$). *gba*^{KO} marked in

stripes as in PCA it clusters with group B (Supplementary Fig.7). (b) Example of hierarchical clustering of parkinsonism mutants based on their scaled behavior data without “comp 1” feature resulting in group 1-5. (c-n) Heatmaps representing the Jaccard similarity index of behavior groups A1 – B2 with groups 1-5 revealed by excluding one feature each. (o) Summary of the Jaccard similarity index for each behavior group A1 – B2 highlights the stability of the originally identified clusters.

Behavior features are independent of the NMF features

To assess if the user-defined behavior parameters and the NMF components are independent, we analysed their correlation for each of the mutants. The linear regression plots (Supplementary Fig.6, Reviewer Fig.9) show very little or no correlation of NMF components and behavior features. This confirms that there is added value to include both user-defined behavior parameters and NMF components in the hierarchical clustering. We included these data in the reworked manuscript.

Reviewer Fig.9 (Supplementary Fig 6): Behavior features do not generally correlate with NMF parameters, related to Fig.2. (a-d') Linear regression plots with 95% confidence intervals (grey area) of behavioral features with NMF components. Dots represent the mean of parkinsonism mutants. R² and p-values are indicated. (e') Summary heatmap of R² values of every behavioral parameter with component 0-4.

The output of ethoscope monitoring is locomotion data

The data we obtain from the ethoscope monitoring is indeed locomotion data (elaborated on in the materials and methods section). Specifically, using the R packages *behavr*, *scopr* and *sleep*³⁹ (v0.3.99) with the behavior annotator, we obtain per fly the location “x” for every interval. Based on these data we computed the behavior features which were interpreted as “sleep” or “awake” as well as activity parameters (e.g. velocity if awake, etc). The “sleep” data in Fig. 2a,b,e,f. are meant as an example of interpretation of these movement data and how they translate into this particular “sleep” traces (as done before by others^{40,41}).

3. Use of ERG to characterize parkinsonism: It is unclear what is(are) the rationale behind choosing ERGs as a readout for mutants that showed primarily dopaminergic impairment (Figure 3). The major neurotransmitter involved in *Drosophila* phototransduction is histamine. The ERG signal is a summation of sustained receptor potentials (depolarization) in photoreceptors and the activity of downstream laminar neurons that are receiving synaptic input from the photoreceptors. Therefore, why was only depolarization change measured and ON and OFF transients excluded? Moreover, due to other factors such as pigment cells that affect the initial depolarization response after light stimulation, measurement of receptor potential (depolarization) is commonly measured when it reaches plateau [Belusic, G., ERG in *Drosophila*, in *Electroretinograms*. 2011, InTechOpen; Vilinsky I, Johnson KG. *Electroretinograms in Drosophila: a robust and genetically accessible electrophysiological system for the undergraduate laboratory*. J Undergrad Neurosci Educ. 2012]. In addition, why was 4 DAE chosen for ERG readout when other experiments were performed in aged flies given the evidence for normal synaptic innervation in young flies (Figure 1)? The traces shown in Figure 3 do not show significant synaptic dysfunction. The authors conclude that some combinations are alleviating, but this measurement is not tracked with age.

Rationale for ERGs as a readout

We are sorry for the confusion. The ERG screen is not meant to model or characterize parkinsonism at this stage of the manuscript. ERGs are commonly and successfully used to test for underlying dysfunction in the circuitry of the visual system. Given that the PD genes we modelled here are broadly expressed (also in visual system neurons of the fly), we used ERG recordings as an orthogonal approach to validate the results of our behavior-based clustering (Fig.2). We opted for ERGs, as they allow for reasonable high-throughput screening (276 gene pairs) while providing at the same time a sensitive and quantitative readout of neuronal dysfunction.

Rationale for depolarisation amplitude

We used the depolarisation amplitude for the genetic interaction analysis as one phenotype we can obtain from these ERG traces, as this was the more robust and less variable value as compared to for example the ‘on’ or ‘off’ peak amplitude.

ERG measurements in young flies and significant synaptic dysfunction

By choosing a young age for the ERG measurements we are testing for molecular dysfunction at a similar age as when we performed the behavior monitoring (Fig.2). As pointed out by the reviewer, Supplementary Fig.2 shows that at the morphological level, the dopaminergic innervation is at this young age unaffected. However, electrophysiological properties might already be defective at this age. Indeed Fig.3b (Reviewer Fig.10b) shows significant changes of the depolarisation amplitude in heterozygous mutant flies at this early age (blue-colored bars, $p < 0.05$, ANOVA Kruskal-Wallis with Benjamini-Hochberg). Among the mutants with a significantly reduced depolarisation amplitude are heterozygous *DJ-1a/b* and *iPLA2-VIA*, for which representative traces are shown in Fig.3c (Reviewer Fig.10c).

Following published methodology to interpret genetic interactions, we consider a genetic interaction only when $|GI \text{ strength}| \geq 0.3$. Gene pairs with scores within -0.3 to $+0.3$ were thus treated as no/weak interaction (revised Material and Method section)^{42,43}. The gene pair of *DJ-1a/b* and *iPLA2-VIA* has a GI strength of 0.36 and is therefore considered as an alleviating genetic interaction.

Furthermore, the Bayesian statistics shows that the high density interval (HDI) of the gene pair of *DJ-1a/b* and *iPLA2-VIA* does not include 0 ($\text{dist}_0 = 0.28$, Supplementary Table 3), which suggest a deviation from the prediction of “no interaction” by 28%. Similarly, the gene pair of the known interactors, *park* and *Pink1*^{44,45}, shows a deviation from the prediction of “no interaction” by 31% ($\text{dist}_0 = 0.31$, Supplementary Table 3). While the differences are sometimes small, we recorded from large groups of animals and applied rigorous statistics to interpret our data. In the reworked manuscript we now also show several additional examples for non-interacting and aggravating gene pair interactions: Reviewer Fig.10c,e,f (Fig.3c-c'') and additional examples in Supplementary Fig.11 (Reviewer Fig.10d,g).

Reviewer Fig.10: Genetic interaction screen supports the assignment of parkinsonism genes to behavior groups. Related to Supplementary Table 3 and Supplementary Fig.11. (a) Scheme of genetic interaction screen using ERGs. The amplitude of the ERG-depolarization of single heterozygous mutants (gene A and B) are normalized to the control ($CS^{w1118}w+$). In the non-interacting model, the expected double heterozygous parkinsonism mutant effect is computed by multiplying the single mutant effects (depolarization amplitude of 50% in this example) and compared to the observed double heterozygous gene pair. Thereby, the GI strength of gene pairs are defined as “aggravating”, “alleviating” or “not interacting”. (b) Normalized depolarization of heterozygous single mutants at 4 ± 1 d after eclosion. Blue

colored bars represent significant decrease compared to control (dark grey), $p < 0.05$, ANOVA Kruskal-Wallis with Benjamini-Hochberg. Bars: mean \pm SD; points represent individual animals $N \geq 10$. (c-g) Representative ERG traces of alleviating, non-interacting and aggravating genetic interactions: the ERG amplitude of double heterozygous mutants of *iPLA2-VIA* and *DJ-1a/b* (c) as well as *rme-8* and *Vham8.9* (d) are rescued, while the ERG amplitude of gene pairs *Gdh* and *Vps13* show no difference to the expected ERG amplitude (e). The double heterozygous mutants of *Vps35* and *omi* (f) as well as the gene pair of *Pink1* and *nutcracker* (g) show an aggravated ERG amplitude compared to the single heterozygous mutants. Black, bold traces show the mean trace, while light grey traces show individual recordings. Dashed lines indicate baseline and 100%, respectively, while the bold lines represent the expected ERG amplitudes. ERG, electroretinogram. GI, genetic interaction.

Classification of genetic interaction independent of age

A genetic interaction is alleviating, if the gene pair shows a higher depolarisation than the depolarisation of the single heterozygous mutants would suggest (expected depolarisation Fig.3a), as outlined in the materials and methods section. Therefore, alleviation or aggravation is not related to an effect at an older age but refers to this particular phenotype under these conditions (and age); we hope the reviewer agrees that it would be practically very difficult to extend this intensive and large experiment to include older age groups.

Moreover, Figure 1 results show significant reduction of climbing performance with age as well as reduction of TH positive area for most of the mutants screened. This data is not consistent with the reduction of depolarization from ERGs in Figure 3. Some mutants such as nutcracker showed significant reduction of TH+ staining and climbing performance but no change in depolarization. As well as others showed no significant reduction in TH positivity or climbing, but significant depolarization. These discrepancies should be discussed in more detail.

As requested, we have added discussion on this in the reworked manuscript. We do want to point out that we do not see these differences as discrepancies but as different phenotypes that emerge in these animals as the result of the presence of PD-relevant mutations.

4. Scaled “velocity” in Method details \diamond Constructing vectors of temporal behavior: If $d = \log(\text{pixel distance})$, using $v = 10^{(d/1000)}/0.0042$, I get only values of $v > 240$, since $1/0.0042$ is the minimum. Is that correct or is there a typo or did I misunderstand? How was this expression for v derived and why was it necessary to re-scale movements? Either way, the 3 speed categories are arbitrary. It is not clear why the authors use rescaled value of velocity (actually speed, if I understand correctly, since velocity is a vector) in Supplementary figure 4 and not the values determined in mm/sec directly from the videos.

We understand the confusion of the reviewer. We used the framework of Geissmann, et al (2017)⁴¹: during the ethoscope monitoring the results are saved as “distances as a fraction of the roi width”, on which a numerical transformation has been applied for storage efficiency taking the \log_{10} and multiplied by 1000 ($xy \text{ dist}_{\log_{10} \times 1000}$). During the offline analysis in R this numerical transformation is undone. Here is the step-by-step calculation:

Python code (runs online, as the experiment progresses)

1. The current position of the fly as x,y coordinates relative to the roi width (so $x = 1$ means fully to the right and $x=0$ fully to the left) is specified:
https://github.com/gilestrolab/ethoscope/blob/bf3208326fd849033f210eac64d30078083d11f2/src/ethoscope/trackers/adaptive_bg_tracker.py#L444
2. The euclidean distance between the new position and the old position (pos) \rightarrow $\text{abs}(pos - \text{self._old_pos})$ is computed.

3. Then the result is multiplied by a 1000 -> $\text{abs}(\text{pos} - \text{self_old_pos}) * 1000$
4. To always have positive results the smallest amount of movement possible (1 pixel) is add -> $1./\text{float}(w_im) + \text{the result of 3.}$
5. Then the log10 is calculated.
https://github.com/gilestrolab/ethoscope/blob/bf3208326fd849033f210eac64d30078083d11f2/src/ethoscope/trackers/adaptive_bg_tracker.py#L445

This result is saved in the db file and is called xy_dist_log10x1000.

Computation in R (offline, after the experiment is finished):

1. dt is computed
https://github.com/rethomics/sleepr/blob/f85bbc56ac7d2f17e2d82352fc549bead3f538f2/R/motion_detectors.R#L44
2. The transformation is undone by reading xy_dist_log10x1000 and retrieving the distance
https://github.com/rethomics/sleepr/blob/f85bbc56ac7d2f17e2d82352fc549bead3f538f2/R/motion_detectors.R#L46
3. the velocity is computed by dividing the distance with dt
https://github.com/rethomics/sleepr/blob/f85bbc56ac7d2f17e2d82352fc549bead3f538f2/R/motion_detectors.R#L47
4. "a" is defined as the velocity coefficient (s. below)
https://github.com/rethomics/sleepr/blob/f85bbc56ac7d2f17e2d82352fc549bead3f538f2/R/motion_detectors.R#L49
5. The distance is calculated by using the velocity (3.) and multiplying it by dt again, which will be divided by "a" (the result is > 1 if the distance is > than the coefficient, and vice versa)
https://github.com/rethomics/sleepr/blob/f85bbc56ac7d2f17e2d82352fc549bead3f538f2/R/motion_detectors.R#L74
6. This ratio sets the threshold to compute whether the fly was moving or not: If the ratio is > 1 the fly was moving.
https://github.com/rethomics/sleepr/blob/f85bbc56ac7d2f17e2d82352fc549bead3f538f2/R/motion_detectors.R#L82

The velocity coefficient $a = 0.0042$ is a variable depending on the structure of the background noise and the types of foreground movements and therefore is empirically determined. It is expressed in units of the ROI width, which is around 60 mm. In other words, if the fly centroid changes from one frame to the next by more than $0.0042 * 60\text{mm} = 0.252\text{ mm}$, then the code computes that the fly has moved. For reference, a fly is roughly 2.5 mm in length, so this is equivalent to 10% of its body length and therefore a reasonable threshold (also used by others).

The 3 speed categories mentioned in the materials and methods section were defined by Geissmann, et al (2017)⁴¹, but were not used in our analysis. We have therefore removed it from the materials and methods section.

Additional (less critical) comments:

1. Fig 1. Panel F second p value seems to have wrong > direction (VAC14).
 We thank the reviewer for drawing our attention to this. There was indeed a typo in Fig.1. This has now been corrected in the revised manuscript.
2. Fig. 2. How do non-parkinsonism flies, such as pdfKO, cluster with the mutants considered in this study?

To investigate how *pdf^{KO}* flies cluster with the fly parkinsonism mutants in our behavior screen, we analysed the NMF-defined phenotypes and the explicit behavioral features for the *pdf^{KO}* flies and re-analysed the hierarchical clustering of parkinsonism mutants including the *pdf^{KO}* (marked in green, Reviewer Fig.11, Supplementary Fig.9). The behavioral features of *pdf^{KO}* flies show similarities with *park^{1/Δ21}*, *Pink1^{KO-WS}* and *Chchd2^{KO-WS}* flies and co-cluster with behavior group B (orange, Reviewer Fig.11b). *pdf^{KO}* flies are characterized by defects in evening and morning anticipation (Fig.2bi and 2bii) and reduced latency of (Fig.2bii), which is reflected in high z-score in comp 0, 1 and 4 as well as a decreased z-score in “latency to longest bout” (Reviewer Fig.11b). Similar, but less severe defects can be observed in *park^{1/Δ21}* and *Pink1^{KO-WS}* mutant flies in our study and also previously reported by our group¹. Interestingly, these circadian defects in *park^{1/Δ21}* and *Pink1^{KO-WS}* have been associated with defects in the production of dense core vesicles containing neuropeptides such as *pdf¹*. This molecular link could therefore (in part) explain the co-clustering of *pdf^{KO}* with *park^{1/Δ21}* and *Pink1^{KO-WS}* in our behavior screen and further supports our conclusion that the behavioral clustering reveals common underlying biology. We have included these results in the reworked manuscript as Supplementary Fig.9.

Reviewer Fig.11 (Supplementary Fig.9): The behavior features of *pdf^{KO}* flies cocluster with behavior group B. (a) Original hierarchical clustering of parkinsonism mutants based on their scaled behavior data, which reveal 2 behavior groups (A in blue, B in orange) with 3 and 2 subgroups each (dark and light shades). (b) Hierarchical clustering of parkinsonism mutants based on their scaled behavior data including the non-parkinsonism mutant *pdf^{KO}* flies (in green).

3. In Fig 3c, what are the horizontal dashed lines meant to signify? Could the authors provide more examples of “alleviation” and add examples of “aggravation” as well (pertaining to major concerns# 3)?

Grey dashed lines indicate base line and 100% depolarisation amplitude of the control, respectively, while the bold lines represent the expected ERG amplitudes to allow for easy comparison of the individual ERG traces. We have clarified this in the figure legend. Moreover, we have included examples

for non-interacting and aggravating gene pairs in the revised Fig.3c-c'' (Reviewer Fig.10c,e,f, above) and additional examples in Supplementary Fig.11 (Reviewer Fig.10d,g, above).

4. How Bayesian analysis is used in ERG analysis is unclear and should be justified better with some intermediate (between raw ERG and final process) results. Why is variance in ERG of particular concern while all other data in this study also come with large variability? Couldn't single ERG recordings not be used, like single fly behavioral timeseries?

In the Supplementary table 3 we have now included the following:

1. We have added the normalized single heterozygous depolarisation amplitude values. From these values the mean was determined and subsequently the mean of the single heterozygous mutants were multiplied to receive the expected values per gene pair.
2. We have added the observed normalized depolarisation for every gene pair.
3. In addition we also shared the code used for the calculations (<https://github.com/verstreckenlab/drosophila-parkinsonism-subgroups>).

We are not using Bayesian analysis for any other analysis, as we are not looking at genetic interaction in the other assays.

We have used Bayesian analysis for the genetic interaction screen, because the expected depolarization value of the double mutant is obtained by multiplying the mean value of each single mutant, with no regard for the variability of each of the single mutant's distribution. This would neglect the variability observed in the single mutants i.e. it would be equivalent to having measured each single mutant only once with a value that happened to be the mean of the collected observations. The ensuing statistics would then be biased, because the variance of the null model (no interaction) would be underestimated, leading to a potentially higher chance of committing statistical errors of Type alpha (wrong rejection of the null hypothesis). With Bayesian analysis, the variance of each single mutant is accessible to the analysis of the difference between expected and observed depolarization values of the double mutant, therefore making less likely to commit the mentioned .

5. Suppl Fig 2. The authors report seizure -like activity. The methodology used to define a seizure was unclear. Is it based on previously published methods? Are the seizures only occurring after a mechanical trigger (vortex)? Do the flies exhibit any involuntary movements? What about after L-Dopa administration? Supplementary video recordings and an expanded description of the assay would be helpful.

We used an established methodology to assess seizure-like behavior of flies¹⁶. We did occasionally observe seizure-like behavior especially while handling mutant flies, and this triggered us to systematically examine this behavior.

For the assay, male flies were transferred in groups of 5 ± 1 into clean vials. After a resting phase of at least 30 min the flies were mechanically stimulated by vortexing the vial for 10 s at maximum intensity. If flies showed involuntary movements while laying on their back or they showed paralysis within 10 s after the mechanical stimulation, they were scored as flies with seizure-like behavior. For the quantification, we show the percentage of flies without seizure-like behavior. We extended the description of this assay in the material and methods section, added the reference and included example video recordings in supplementary information (Supplementary videos 1-6).

In response to the reviewer's request, we also treated mutants that showed seizure-like behavior with L-/D-Dopa in the 10 days prior to conducting the seizure-like behavior assay (same feeding paradigm as for the SING assay with L-/D-Dopa treatment). The seizure-like behavior of *Rab39^{KO-WS}*, *iPLA2-VIA^{KO-WS}*, *Gdh^{KO-WS/+}* and *gba^{KO}* was not rescued by the L-Dopa treatment (Supplementary Fig. 2e and Reviewer Fig.12e). We included these new data in the manuscript.

Reviewer Fig.12 (Supplementary Fig.3): Several parkinsonism mutants show seizure-like behavior, related to Fig.1. (a,b) Quantification of seizure-like behavior in parkinsonism mutants after sensory stimulation (vortex) at (a) 6±1 d (young) and (b) 26 ±2 d after eclosion (except for *nutcracker*^{KO-WS} at 15±1 d and, *Pink1*^{KO-WS} at 22±2 d). Orange colored bars represent p<0.01, ANOVA Kruskal-Wallis with Benjamini-Hochberg compared. Bars: mean ± SD; points are groups of animals and N≥9. (c) Representative images of 25±1 d old control and *Rab39*^{KO-WS} after stimulation, black arrows indicate flies with seizure-like phenotypes. (d) Venn diagram representing the overlap of patients with familial forms of parkinsonism described to suffer from seizures and fly mutants with seizure-like behavior. (e) Quantification of a subset of parkinsonism mutants with seizure-like behavior after sensory stimulation (vortex) at 26 ±2 d after eclosion treated with L-Dopa or D-Dopa (control, grey bars) in the 10 d prior to the assay. Two-way ANOVA with Tukey's multiple comparison, significant p values are indicated with * p<0.05, ** p<0.01, *** p<0.001, **** p<0.0001. Bars: mean ± SD; points are groups of animals and N≥8.

6. Suppl Fig. 6. By looking at gene expression profiles in the brain, the authors conclude that clustering of gene expression cannot explain clustering of mutants according to behavior. Since PD-important genes are likely expressed widely in the fly, can the authors comment on the possibility of linking genes

to behavior by examining expressions outside the brain as well? For instance, the gut (and gut-brain connection) has received particular attention in several Parkinson's studies. This is a great suggestion and to investigate whether a shared expression profile in cell types outside the brain could explain the clustering of fly parkinsonism genes identified in the behavior screen, we performed expression profile analysis and hierarchical clustering of fly parkinsonism genes across all identified cell types of the fly body. Similar to assessing expression across cell types in the brain only, the resulting gene expression clusters when assessing gene expression across the fly body do not overlap with the behavior-based subgroups (Supplementary Fig.10c and Reviewer Fig.13). These data were also included in the reworked paper.

Reviewer Fig.13 (Supplementary Fig.10): Parkinsonism gene expression clustering across cell-types of the fly brain and body. (a) Heatmap and hierarchical clustering of scaled parkinsonism gene expression across all identified cell types in single-cell RNAseq atlas⁴⁶ of brains of young (5±1 d) control flies. Parkinsonism genes are color coded according to behavior groups. (b) Example UMAPs of *Pink1*, *park*, *Chchd2*, *nutcracker*, *iPLA2-VIA*, *Rab39*, *gba1a/b* and *DJ-1a/b* normalized and logarithmized expression values. (c) Heatmap and hierarchical clustering of scaled parkinsonism gene expression across all identified cell types of the fly body in single-cell RNAseq atlas⁴⁷. Parkinsonism genes are color coded according to behavior groups.

Reviewer #5 (Remarks to the Author):

References

1. Valadas, J. S. *et al.* ER Lipid Defects in Neuropeptidergic Neurons Impair Sleep Patterns in Parkinson's Disease. *Neuron* **98**, 1155-1169.e6 (2018).
2. Gruntz, K. *et al.* Parkinson disease and the risk of epileptic seizures. *Ann Neurol* **83**, 363–374 (2018).
3. Ben Romdhan, S. *et al.* A Novel SYNJ1 Mutation in a Tunisian Family with Juvenile Parkinson's Disease Associated with Epilepsy. *J Mol Neurosci* **66**, 273–278 (2018).
4. Ng, J. *et al.* *DNAJC6* Mutations Disrupt Dopamine Homeostasis in Juvenile PARKINSONISM-DYSTONIA. *Movement Disorders* **35**, 1357–1368 (2020).
5. Taipa, R. *et al.* DJ-1 linked parkinsonism (PARK7) is associated with Lewy body pathology. *Brain* **139**, 1680–1687 (2016).
6. Ramser, J. *et al.* A unique exonic splice enhancer mutation in a family with X-linked mental retardation and epilepsy points to a novel role of the renin receptor. *Human Molecular Genetics* **14**, 1019–1027 (2005).
7. Mata, I. F. *et al.* The RAB39B p.G192R mutation causes X-linked dominant Parkinson's disease. *Mol Neurodegeneration* **10**, 50 (2015).
8. Karkheiran, S., Shahidi, G. A., Walker, R. H. & Paisan-Ruiz, C. PLA2G6-associated Dystonia–Parkinsonism: Case Report and Literature Review. *Tremor and Other Hyperkinetic Movements* Tremor and Other Hyperkinetic Movements (2015) doi:10.7916/D84Q7T4W.
9. Sidransky, E. & Lopez, G. The link between the GBA gene and parkinsonism. *The Lancet Neurology* **11**, 986–998 (2012).
10. Pavlidis, P. & Tanouye, M. Seizures and failures in the giant fiber pathway of *Drosophila* bang-sensitive paralytic mutants. *J. Neurosci.* **15**, 5810–5819 (1995).
11. Stilwell, G. E., Saraswati, S., Littleton, J. T. & Chouinard, S. W. Development of a *Drosophila* seizure model for *in vivo* high-throughput drug screening. *Eur J of Neuroscience* **24**, 2211–2222 (2006).
12. Marley, R. & Baines, R. A. Increased persistent Na⁺ current contributes to seizure in the slamdance bang-sensitive *Drosophila* mutant. *Journal of Neurophysiology* **106**, 18–29 (2011).
13. Lee, J. & Wu, C.-F. Electroconvulsive seizure behavior in *Drosophila*: analysis of the physiological repertoire underlying a stereotyped action pattern in bang-sensitive mutants. *J Neurosci* **22**, 11065–11079 (2002).

14. Burg, M. G. & Wu, C.-F. Mechanical and temperature stressor-induced seizure-and-paralysis behaviors in *Drosophila bang*-sensitive mutants. *J Neurogenet* **26**, 189–197 (2012).
15. Parker, L., Padilla, M., Du, Y., Dong, K. & Tanouye, M. A. *Drosophila* as a Model for Epilepsy: *bss1s* a Gain-of-Function Mutation in the Para Sodium Channel Gene That Leads to Seizures. *Genetics* **187**, 523–534 (2011).
16. Kuebler, D. & Tanouye, M. A. Modifications of seizure susceptibility in *Drosophila*. *J Neurophysiol* **83**, 998–1009 (2000).
17. Reynolds, E. R. *et al.* Treatment with the antiepileptic drugs phenytoin and gabapentin ameliorates seizure and paralysis of *Drosophila bang*-sensitive mutants. *J. Neurobiol.* **58**, 503–513 (2004).
18. Byers, N., Hahm, E.-T. & Tsunoda, S. Slo2/K_{Na} Channels in *Drosophila* Protect against Spontaneous and Induced Seizure-like Behavior Associated with an Increased Persistent Na⁺ Current. *J. Neurosci.* **41**, 9047–9063 (2021).
19. Streit, A. K., Fan, Y. N., Masullo, L. & Baines, R. A. Calcium Imaging of Neuronal Activity in *Drosophila* Can Identify Anticonvulsive Compounds. *PLoS ONE* **11**, e0148461 (2016).
20. Housden, B. E., Terriente-Felix, A. & Bray, S. J. Context-dependent enhancer selection confers alternate modes of notch regulation on *argos*. *Mol Cell Biol* **34**, 664–672 (2014).
21. Riemensperger, T. *et al.* A single dopamine pathway underlies progressive locomotor deficits in a *Drosophila* model of Parkinson disease. *Cell Rep* **5**, 952–960 (2013).
22. Sun, J. *et al.* Neural Control of Startle-Induced Locomotion by the Mushroom Bodies and Associated Neurons in *Drosophila*. *Front Syst Neurosci* **12**, 6 (2018).
23. Hendricks, J. C. *et al.* Rest in *Drosophila* Is a Sleep-like State. *Neuron* **25**, 129–138 (2000).
24. Keene, A. C. & Duboue, E. R. The origins and evolution of sleep. *J Exp Biol* **221**, jeb159533 (2018).
25. Dodet, P. *et al.* Sleep disorders in Parkinson's disease, an early and multiple problem. *npj Parkinsons Dis.* **10**, 46 (2024).
26. Kasten, M. *et al.* Nonmotor Symptoms in Genetic Parkinson Disease. *Arch Neurol* **67**, (2010).
27. Pesah, Y. *et al.* *Drosophila parkin* mutants have decreased mass and cell size and increased sensitivity to oxygen radical stress. *Development* **131**, 2183–2194 (2004).
28. Cha, G.-H. *et al.* Parkin negatively regulates JNK pathway in the dopaminergic neurons of *Drosophila*. *Proc. Natl. Acad. Sci. U.S.A.* **102**, 10345–10350 (2005).
29. Srinivasan, E. *et al.* Alpha-Synuclein Aggregation in Parkinson's Disease. *Front Med (Lausanne)* **8**, 736978 (2021).
30. Schneider, S. A. & Alcalay, R. N. Neuropathology of genetic synucleinopathies with parkinsonism: Review of the literature. *Mov Disord* **32**, 1504–1523 (2017).
31. Klein, C. & Westenberger, A. Genetics of Parkinson's Disease. *Cold Spring Harbor Perspectives in Medicine* **2**, a008888–a008888 (2012).
32. Caliński, T. & Harabasz, J. A dendrite method for cluster analysis. *Communications in Statistics* **3**, 1–27 (1974).
33. Player Vectors: Characterizing Soccer Players' Playing Style from Match Event Streams. in *Lecture Notes in Computer Science* 569–584 (Springer International Publishing, Cham, 2020). doi:10.1007/978-3-030-46133-1_34.
34. Guven, E. Decision of the Optimal Rank of a Nonnegative Matrix Factorization Model for Gene Expression Data Sets Utilizing the Unit Invariant Knee Method: Development and Evaluation of the Elbow Method for Rank Selection. *JMIR Bioinform Biotech* **4**, e43665 (2023).

35. Aggarwal, C. C. *Data Mining: The Textbook*. (Springer International Publishing, Cham, 2015). doi:10.1007/978-3-319-14142-8.
36. Arshad, Z. & McDonald, J. F. Changes in gene-gene interactions associated with cancer onset and progression are largely independent of changes in gene expression. *iScience* **24**, 103522 (2021).
37. Galindo-Hernández, R., Rodríguez-Vázquez, K., Galán-Vásquez, E. & Hernández Castellanos, C. I. Online-adjusted evolutionary biclustering algorithm to identify significant modules in gene expression data. *Briefings in Bioinformatics* **26**, bbae681 (2024).
38. de Reuver, S. *et al.* Genetic overlap between idiopathic scoliosis and schizophrenia in the general population. *Spine Deform* **13**, 413–422 (2025).
39. Geissmann, Q., Garcia Rodriguez, L., Beckwith, E. J. & Gilestro, G. F. Rethomics: An R framework to analyse high-throughput behavioural data. *PLoS ONE* **14**, e0209331 (2019).
40. Geissmann, Q., Beckwith, E. J. & Gilestro, G. F. *Most Sleep Does Not Serve a Vital Function. Evidence from Drosophila Melanogaster*. <http://biorxiv.org/lookup/doi/10.1101/361667> (2018) doi:10.1101/361667.
41. Geissmann, Q. *et al.* Ethoscopes: An open platform for high-throughput ethomics. *PLoS Biol* **15**, e2003026 (2017).
42. Shi, X. *et al.* Combinatorial GxGxE CRISPR screen identifies SLC25A39 in mitochondrial glutathione transport linking iron homeostasis to OXPHOS. *Nat Commun* **13**, 2483 (2022).
43. Heigwer, F. *et al.* Time-resolved mapping of genetic interactions to model rewiring of signaling pathways. *Elife* **7**, e40174 (2018).
44. Clark, I. E. *et al.* *Drosophila pink1* is required for mitochondrial function and interacts genetically with parkin. *Nature* **441**, 1162–1166 (2006).
45. Park, J. *et al.* Mitochondrial dysfunction in *Drosophila* PINK1 mutants is complemented by parkin. *Nature* **441**, 1157–1161 (2006).
46. Pech, U. *et al.* Synaptic deregulation of cholinergic projection neurons causes olfactory dysfunction across 5 fly Parkinsonism models. Preprint at <https://doi.org/10.1101/2023.03.11.532176> (2023).
47. Li, H. *et al.* Fly Cell Atlas: A single-nucleus transcriptomic atlas of the adult fruit fly. *Science* **375**, eabk2432 (2022).

Point-by-point response to REVIEWERS' COMMENTS

Reviewer #1 (Remarks to the Author):

The revisions, including new experimental data, have resulted in a much improved manuscript. Collectively, a remarkable amount of work is presented, making for an important and valuable contribution. It appears that a technical glitch remains with accessing Supplemental Table 3.

We thank the reviewer for this very positive assessment and for recognizing the additional experimental work and the resulting improvements to the manuscript.

Reviewer #2 (Remarks to the Author):

This manuscript provides functional analyses of mutations in *Drosophila* homologs of 24 genes that have been linked to Parkinson and other motor dysfunctions in humans. They use multiple assays to cluster the phenotypes of these mutants and show that the genes fall into two broad groups (or 3 groups depending on how one sets the cutoffs). Remarkably, orthogonal assays lead to the same basic groupings; even more remarkably, response to pharmacological treatment largely obeys the same groupings. Perhaps most striking, this is true not only of genes that are known to have related functions, but even of genes that seem superficially to have little in common but that co-cluster in the unbiased assays and then are rescued by the same drugs.

This paper brings an unexpected and long-sought order to what has been a bewildering multiplicity of functions and properties of PD-associated genes. It also offers a whole new way to think about the basic pathogenesis of PD, a meaningful way to segregate pathological genes into functional pathways, and a model for how we can dissect other diseases with complex genetic etiology, including, AD, FTD-ALS, and others. It is of huge significance to the specific field of Parkinson disease, to the larger field of human neurodegenerative disease, and in the broadest sense as a model for dissection of complex genetic processes.

I found the original submission of this manuscript to be quite impressive; the revised version is even better. In my opinion the authors are to be applauded for the clarity and thoroughness with which they addressed the comments of all the reviewers, myself included. I am entirely satisfied by their responses. There were a number of referee requests to expand on details of the clustering methods, which clearly were central to the authors' conclusions; I found the responses to be sensible and convincing.

One thing that struck me in reading the comments is that a number of the referee concerns boil down to the question of why assays like ERG and sleep, that seem to be only peripherally related to diagnostic phenotypes of "parkinsonism", should be appropriate tools for clustering molecular lesions associated with motor dysfunction. However, it seems to me that the success of these measures (as demonstrated by the concordance between clustering by multiple assays and by

response to treatment) is one of the most profound results of the entire project. It provides an astonishing demonstration that the disease pathogenesis is rooted, not in the details of dopamine biology per se, but in basic mechanisms of neuronal cell biology. These data provide powerful support for the idea that it is the selective sensitivity of particular neuronal cell types to specific aspects of that basic cell biology that distinguish among the various neurodegenerative diseases (an idea that is also underscored by results in a previous study from this group, Praschberger, 2023). The work by these authors now gives us a way to approach the basis of that selective sensitivity that has been so elusive.

We are very grateful to the reviewer for their exceptionally positive and detailed assessment of the manuscript. We appreciate the recognition of the novelty, clarity, and potential impact of our work.

Reviewer #3 (Remarks to the Author):

I am fully satisfied with the authors' responses and additional experiments, and I have no further questions. The authors have appropriately addressed the reviewers' questions and critiques, and this revision is sufficient for publication.

We appreciate the reviewer's positive feedback and recommendation for publication.

Reviewer #4 (Remarks to the Author):

The investigators have provided a substantially improved revised manuscript. The changes to Figure 3 and its related discussion offer greater clarity regarding both the methodological approach and the conclusions drawn regarding the genetic interaction screen using ERGs. Inclusion of the Calinski-Harabasz index bolsters their earlier claims about data clustering. Revised Supplementary Figures 6 and 8 are helpful. The revised Supplementary Figure 3 along with newly included videos capturing seizure-like behavior is a valuable addition, that enables a more comprehensive assessment of the described phenotype. Moreover, the incorporation of small molecule treatment for all mutants, as detailed in revised Figure 4, provided compelling evidence to support the assigned classifications. I still find description of fly speed calculations unnecessarily complicated but that may be more due to the work of Geissmann et al rather than the current manuscript.

The revised manuscript is recommended for publication.

We sincerely thank the reviewer for their thorough and highly positive evaluation of the revised manuscript and for recognizing the added clarity and strength provided by the revised figures, supplementary materials, and additional analyses. We appreciate the recommendation for publication.

Reviewer #5 (Remarks to the Author):
